# Learning to Reason under Off-Policy Guidance

**Jianhao Yan**[123]* **Yafu Li**[2]*† **Zican Hu**[42] **Zhi Wang**[42] **Ganqu Cui**[2] **Xiaoye Qu**[2]
**Yu Cheng**[5]† **Yue Zhang**[3]†

[1] Zhejiang University [2] Shanghai AI Laboratory [3] Westlake University [4] Nanjing University
[5] The Chinese University of Hong Kong
Corresponding to: `chengyu@cse.cuhk.edu.hk`, `yue.zhang@wias.org.cn`

## Abstract

Recent advances in large reasoning models (LRMs) demonstrate that sophisticated behaviors such as multi-step reasoning and self-reflection can emerge via reinforcement learning with verifiable rewards (*RLVR*). However, existing *RLVR* approaches are inherently "on-policy", limiting learning to a model's own outputs and failing to acquire reasoning abilities beyond its initial capabilities. To address this issue, we introduce **LUFFY** (**L**earning to reason **U**nder o**FF**-polic**Y** guidance), a framework that augments *RLVR* with off-policy reasoning traces. LUFFY dynamically balances imitation and exploration by combining off-policy demonstrations with on-policy rollouts during training. Specifically, LUFFY combines the Mixed-Policy GRPO framework, which has a theoretically guaranteed convergence rate, alongside policy shaping via regularized importance sampling to avoid superficial and rigid imitation during mixed-policy training. Compared with previous RLVR methods, LUFFY achieves an over **+6.4** average gain across six math benchmarks and an advantage of over **+6.2** points in out-of-distribution tasks. Most significantly, we show that LUFFY successfully trains weak models in scenarios where on-policy RLVR completely fails. These results provide compelling evidence that LUFFY transcends the fundamental limitations of on-policy RLVR and demonstrates the great potential of utilizing off-policy guidance in RLVR.

## 1 Introduction

Recent breakthroughs in large reasoning models, including OpenAI-o1 [1], DeepSeek-R1 [2], and Kimi-1.5 [3], have demonstrated remarkable capabilities in complex reasoning tasks. These models have shown unprecedented proficiency in generating extensive Chains-of-Thought (CoT, [4]) responses and exhibiting sophisticated behaviors, such as self-reflection and self-correction. Particularly noteworthy is how these achievements have been realized through *reinforcement learning with purely verifiable rewards* (RLVR), as demonstrated by recent efforts including Deepseek R1 [2, 5, 6, 7]. The emergence of long CoT reasoning and self-reflection capabilities through such straightforward reward mechanisms, termed the "aha moment", represents a significant advancement in the field.

Nevertheless, the reinforcement learning methods behind the success have a fundamental limitation worth highlighting: it is inherently on-policy, constraining learning exclusively to the model's self-generated outputs through iterative trials and feedback cycles. Despite showing promising results, on-policy RL is bounded by the base LLM itself [8, 9]. In essence, reinforcement learning under this setting amplifies existing behaviors rather than introducing genuinely novel cognitive capacities. Recent study [10] corroborates this constraint, demonstrating that models like Llama 3.2 [11] quickly reach performance plateaus under RL training precisely because they lack certain foundational

---

* Co-first authors. Work was done during Jianhao Yan's internship at Shanghai AI Laboratory. Yafu Li is the Project Lead.
† Corresponding authors.

39th Conference on Neural Information Processing Systems (NeurIPS 2025).

cognitive behaviors necessary for further advancement. This inherent limitation provokes critical questions about the effectiveness and scope of RL for reasoning: *How can we empower LLMs to acquire reasoning behaviors surpassing their initial cognitive boundaries?*

In this paper, we introduce **LUFFY**: **L**earning to reason **U**nder o**FF**-polic**Y** guidance, addressing this issue by introducing external guidance from a stronger policy (e.g., from DeepSeek-R1). The strong policy serves as guidance for diverging the training trajectory beyond the limitations of the model's initial capabilities. Unlike on-policy RL, where the model can only learn from its own generations, our approach leverages off-policy learning to expose the model to reasoning patterns and cognitive structures that might otherwise remain inaccessible. This external guidance functions as a form of cognitive scaffolding, allowing the model to observe and internalize reasoning strategies from a more capable teacher model, thereby expanding its reasoning repertoire beyond what self-improvement alone could achieve.

In particular, LUFFY extends on GRPO [12] to *Mixed-Policy GRPO* by introducing a new off-policy objective with importance sampling to calibrate policy gradient, and combining off-policy reasoning traces with models' on-policy roll-outs during advantage computation, as illustrated in Figure 1. Intuitively, since off-policy traces consistently obtain positive rewards, LUFFY enables the model to selectively imitate these high-quality reasoning traces when its own roll-outs fail to achieve correctness, while preserving the capacity for self-driven exploration whenever its generated reasoning steps are successful. In this way, LUFFY achieves a dynamic and adaptive equilibrium between imitation and exploration. To avoid overly rapid convergence and entropy collapse, causing the model to latch onto superficial patterns rather than acquiring genuine reasoning capabilities, we further introduce *policy shaping via regularized importance sampling*, which amplifies learning signals for low-probability yet crucial actions under off-policy guidance. This mechanism encourages the model to preserve exploration throughout training, ultimately enabling it to internalize deeper and more generalizable reasoning behaviors.

LUFFY achieves significant improvements of **+6.4** points on average compared with previous RLVR methods, across AIME24/25 [13], AMC [13], OlympiadBench [14], Minerva [15], and MATH-500 [16] benchmarks, establishing the effectiveness of off-policy learning in RLVR paradigms. Moreover, LUFFY demonstrates superior generalization capability, i.e., an advantage of over **+6.2** points on average, on out-of-distribution tasks, where other off-policy methods fall short. Critically, we show that LUFFY successfully trains weak foundation models, i.e., LLaMA3.1-8B, while on-policy RL fails, providing evidence of LUFFY transcending the limitation of model capacity. Our in-depth analyses demonstrate that LUFFY encourages the model to imitate high-quality reasoning traces while maintaining exploration of its own sampling space, resulting in more reliable and generalizable reasoning capabilities.

Our contributions can be summarized as:

- We introduce **LUFFY**, an approach that incorporates off-policy guidance into GRPO and integrates policy shaping through regularized importance sampling to address entropy collapse, effectively transcending the limitations of on-policy RL. (Sec. 3)
- We empirically demonstrate LUFFY's effectiveness across various foundation models, achieving an average gain of **+6.4** points across six math benchmarks and **+6.2** points on out-of-distribution tasks against previous RLVR methods, establishing a new *state-of-the-art* on RLVR with Qwen2.5-Math-7B. (Sec. 5.1)
- We demonstrate that LUFFY *successfully* trains weaker foundation models where On-Policy RL *fails*. Specifically, while On-Policy RL can only train Llama3.1-8B on simplified datasets, LUFFY effectively trains these models across varying difficulty levels, overcoming capability-based limitations (Sec. 5.2).

## 2 Reinforcement Learning with Verifiable Rewards

**Verifiable Reward Function.** The verifiable reward emphasizes the comparison between the extracted answer from the models' output and the predefined golden answer. For instance, the model is instructed to output the final answer in a certain format, e.g., \boxed{}, and a regex function is used to extract the answer from \boxed{}. Formally, given a model's output $\tau$ to question $q$, the

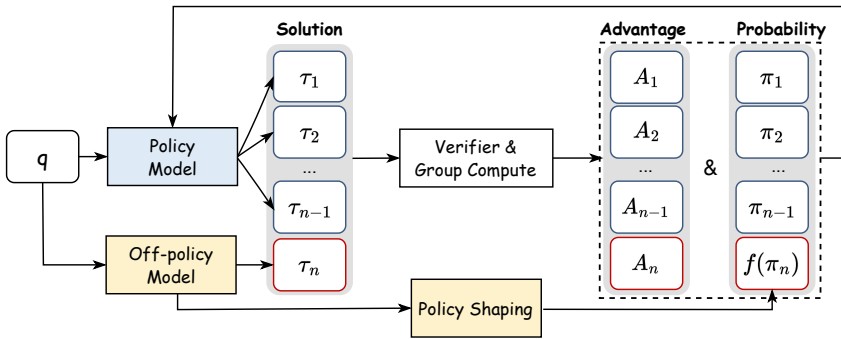

Figure 1: Overview: LUFFY integrates off-policy reasoning traces into reinforcement learning by combining them with on-policy rollouts. Policy shaping emphasizes low-probability but crucial actions, enabling a balance between imitation and exploration for more generalizable reasoning.

reward is defined by,

$$R(\tau) = \begin{cases} 1 & \text{if } \tau \text{ outputs the correct final answer to } q \\ 0 & \text{otherwise} \end{cases} \tag{1}$$

This reward design avoids the risk of reward hacking [17, 18, 19] to a great extent and thus leads to successful scaling of RL training [2].

**Group Relative Policy Optimization (GRPO).** GRPO [12] shows exceptional performance in various tasks, especially to enable effective scaling within the RLVR paradigm [2, 20, 6]. It uses the reward scores of $N$ sampled solutions from a query to estimate the advantage and thus remove the need for an additional value model. Formally, we denote the policy model before and after the update as $\pi_{\theta_{\text{old}}}$ and $\pi_\theta$, where both represent probability distributions over possible actions/tokens at each position. Given a question $q$, a set of sampled solutions $\tau_i$ generated by $\pi_{\theta_{\text{old}}}$, and the reward function $R(\cdot)$, the advantage $A_i$ of each in GRPO is computed by normalized rewards inside the group,

$$A_i = \frac{R(\tau_i) - \text{mean}(\{R(\tau_i) \mid \tau_i \sim \pi_{\theta_{\text{old}}}(\tau), i = 1, 2, \ldots, N\})}{\text{std}(\{R(\tau_i) \mid \tau_i \sim \pi_{\theta_{\text{old}}}(\tau), i = 1, 2, \ldots, N\})}, \tag{2}$$

Then, the RL objective is inherited from the clipped RL objective proposed by PPO [21],

$$\mathcal{J}_{\text{GRPO}}(\theta) = \frac{1}{\sum_{i=1}^N |\tau_i|} \sum_{i=1}^N \sum_{t=1}^{|\tau_i|} \text{CLIP}(r_{i,t}(\theta), A_i, \epsilon) - \beta \cdot \mathbb{D}_{\text{KL}}[\pi_\theta \| \pi_{\text{ref}}]. \tag{3}$$

where $r_{i,t}(\theta) = \pi_\theta(\tau_{i,t}|q, \tau_{i,<t})/\pi_{\theta_{\text{old}}}(\tau_{i,t}|q, \tau_{i,<t})$ is a importance sampling term to calibrate the gradient based on policy gradient theory [22], as the solutions are generated by $\pi_{\theta_{\text{old}}}$ instead of $\pi_\theta$.

The KL divergence $\mathbb{D}_{\text{KL}}$ and the clipped surrogate objective $\text{CLIP}(r, A, \epsilon) = \min[r \cdot A, \text{clip}(r; 1 - \epsilon, 1 + \epsilon) \cdot A]$ empirically ensures that the current policy $\pi_\theta$ is within the trust region [23] of the old policy $\pi_{\theta_{\text{old}}}$. We loosely categorize this method as *On-Policy RL*, indicating that the model is optimized using samples drawn from distributions closely aligned with its current policy. Nevertheless, recent practices [24, 25, 7] have increasingly omitted the KL divergence term, making these methods somewhat less "On-Policy".

## 3 Learning to Reason under Off-Policy Guidance

To facilitate exploration beyond the model's own capabilities, we incorporate *off-policy guidance*, i.e., off-the-shelf reasoning trajectories generated by a stronger reasoning model such as Deepseek R1, into the RLVR learning. We expect the model to learn generalizable knowledge from off-policy beyond superficial imitation and maintain effective and efficient exploration as in RLVR training. In the following sections, we introduce **LUFFY**: **mixed-policy GRPO** (§ 3.1) with **policy shaping** (§3.2). The former one adaptively integrates off-policy trajectories into advantage estimation while the latter encourages continuous exploration throughout training.

### 3.1 Mixed-Policy GRPO

We incorporate off-policy rollouts in GRPO by adding them directly to the group of on-policy rollouts generated by the model itself. Provided an off-policy distribution $\pi_\phi$, this would affect the advantage computations in the following way,

$$\hat{A}_i = \frac{R(\tau_i) - \text{mean}(\mathcal{G}_{\text{on}} \cup \mathcal{G}_{\text{off}})}{\text{std}(\mathcal{G}_{\text{on}} \cup \mathcal{G}_{\text{off}})}, \tag{4}$$

where $\mathcal{G}_{\text{on}} = \{R(\tau_i) \mid \tau_i \sim \pi_{\theta_{\text{old}}}(\tau), i = 1, 2, \ldots, N_{\text{on}}\}$ and $\mathcal{G}_{\text{off}} = \{R(\tau_j) \mid \tau_j \sim \pi_\phi(\tau), j = 1, 2, \ldots, N_{\text{off}}\}$. As the quality of off-policy rollouts is high (yielding high rewards), this group computation naturally assigns higher advantage to off-policy rollouts when the model struggles to generate correct solutions independently. Once the model begins producing successful reasoning traces, on-policy rollouts take precedence, thereby encouraging self-driven exploration.

Extending from the GRPO objective (Eq. 3), we introduce the off-policy objective to incorporate off-policy rollouts. Similar to GRPO and PPO [21, 12], we introduce an importance sampling term $\hat{r}_{j,t}(\theta, \phi) = \pi_\theta(\tau_{j,t}|q, \tau_{j,<t})/\pi_\phi(\tau_{j,t}|q, \tau_{j,<t})$ to calibrate gradient estimates [22]. We refer to this approach as *Mixed-Policy GRPO*:

$$\mathcal{J}_{\text{Mixed}}(\theta) = \frac{1}{Z} \Big( \underbrace{\sum_{j=1}^{N_{\text{off}}} \sum_{t=1}^{|\tau_j|} \text{CLIP}(\hat{r}_{j,t}(\theta, \phi), \hat{A}_j, \epsilon)}_{\text{off-policy objective}} + \underbrace{\sum_{i=1}^{N_{\text{on}}} \sum_{t=1}^{|\tau_i|} \text{CLIP}(r_{i,t}(\theta), \hat{A}_i, \epsilon)}_{\text{on-policy objective}}, \tag{5}$$

where $\hat{r}_{j,t}(\theta, \phi) = \pi_\theta(\tau_{j,t}|q, \tau_{j,<t})/\pi_\phi(\tau_{j,t}|q, \tau_{j,<t})$ and $r_{i,t}(\theta) = \pi_\theta(\tau_{i,t}|q, \tau_{i,<t})/\pi_{\theta_{\text{old}}}(\tau_{i,t}|q, \tau_{i,<t})$. $Z = \sum_{j=1}^{N_{\text{off}}} |\tau_j| + \sum_{i=1}^{N_{\text{on}}} |\tau_i|$ is the normalization factor.

The newly introduced importance sampling term $\hat{r}_{j,t}$ contrasts with the importance sampling term $r_{i,t}$ used in on-policy RL (Eq. 3), where the denominator corresponds to the pre-update roll-out model policy $\pi_{\theta_{old}}$. Since the divergence between $\pi_\theta$ and $\pi_{\theta_{old}}$ is typically much smaller than that between $\pi_\theta$ and the off-policy policy $\pi_\phi$, the off-policy importance sampling ratio $\hat{r}_{j,t}$ tends to be smaller, serving to calibrate gradient estimates from a distinct distribution.

Based on the theoretical analysis of stochastic gradient descent in nonconvex optimization [26], we give a convergence analysis in Theorem 1 to show that our importance-weighted policy gradient estimator in Eq. (5) stabilizes and converges to a stationary point, and the convergence rate is $O(1/\sqrt{K})$, where $K$ is the total number of iterations. The proof can be found in Appendix B.1.

**Theorem 1.** *Suppose the objective function of the policy gradient algorithm $J \in \mathcal{J}_n$, where $\mathcal{J}_n$ is the class of finite-sum Lipschitz smooth functions, has $\sigma$-bounded gradients, and the importance weight $w = \pi_\theta/\pi_\phi$ is clipped to be bounded by $[\underline{w}, \overline{w}]$. Let $\alpha_k = \alpha = c/\sqrt{K}$ where $c = \sqrt{\frac{2(J(\boldsymbol{\theta}^*) - J(\boldsymbol{\theta}^0))}{L\sigma^2 \underline{w}\overline{w}}}$, and $\boldsymbol{\theta}^*$ is an optimal solution. Then, the iterates of our algorithm in Eq. (5) satisfy:*

$$\min_{0 \le k \le K-1} \mathbb{E}[||\nabla J(\boldsymbol{\theta}^k)||^2] \le \sqrt{\frac{2(J(\boldsymbol{\theta}^*) - J(\boldsymbol{\theta}^0))L\overline{w}}{K\underline{w}}} \sigma.$$

The importance sampling ratio in off-policy learning typically involves $\pi_\phi$, representing the behavior policy's probability in off-policy trajectories [27]. Theoretically, our derivations and guarantees hold for any well-defined $\pi_\phi$ distribution. In practice, to facilitate direct integration of high-quality demonstrations from large, powerful models (e.g., DeepSeek-R1), we adopt $\pi_\phi = 1$ for computational efficiency. This practical choice not only avoids the complexity caused by *different tokenization* between the on-policy and off-policy models. It also facilitates the easy incorporation of *off-the-shelf datasets* without recomputation of $\pi_\phi$, as well as preserving theoretical guarantees. We omit the clip operation for the off-policy rollouts, as the clip operation will be imbalanced when $\pi_\phi = 1$.

### 3.2 Policy Shaping via Regularized Importance Sampling

While Mixed-Policy GRPO incorporates off-policy guidance successfully via group computation and importance sampling, a new practical challenge emerges: Mixed-Policy GRPO accelerates

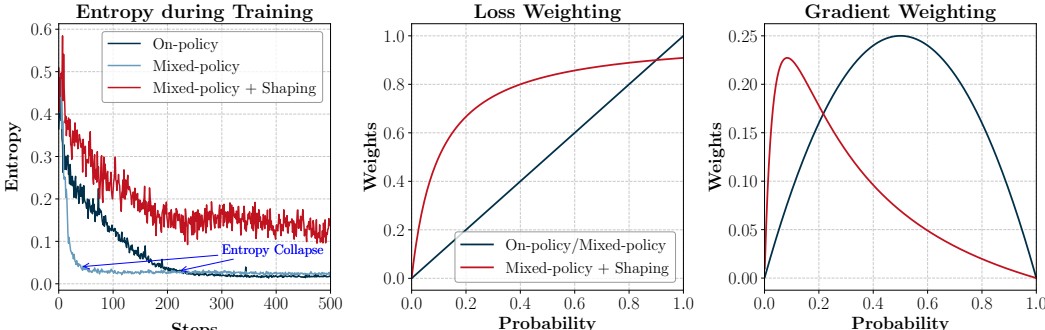

Figure 2: **Left**: generation entropy during training; **Middle**: shaping function value w.r.t. action probability; **Right**: gradient weights w.r.t. action probability.

convergence but significantly reduces exploration (Figure 2, left). Specifically, entropy collapses much faster than in on-policy RL, indicating increasingly deterministic rollouts and a diminished capacity for exploring diverse reasoning trajectories.

This originates from the "hacking" of the Mixed-Policy objective. When combining both learning off-policy and on-policy signals, the model tends to quickly converge toward reinforcing off-policy tokens that are also likely in the on-policy $\pi_\theta$ distribution, and ignoring off-policy tokens that are deviated from the model's original policy, i.e., low-probability tokens that may represent essential reasoning capabilities the model has yet to acquire. We empirically analyze this issue in detail in Section 5.3.

To address this issue, we introduce *policy shaping via regularized importance sampling*, a technique that re-weights the gradient of off-policy distributions to enhance learning from low-probability tokens. In particular, our approach replaces the importance sampling ratio $\hat{r}_{j,t}(\theta, \phi)$ with $f(\hat{r}_{j,t}(\theta, \phi))$, where $f(\cdot)$ represents a transformation function that alters the dynamics between off-policy and on-policy distributions, thereby increasing gradient emphasis on tokens with low probability in the model's standard distribution. Recall that we omit the clip operation for the off-policy rollouts. The loss function with policy shaping can be written as below:

$$\mathcal{J}_{\text{SHAPING}}(\theta) = \frac{1}{Z}\left(\sum_{j=1}^{N_{\text{off}}} \sum_{t=1}^{|\tau_j|} f(\hat{r}_{j,t}(\theta, \phi)) \cdot \hat{A}_j + \sum_{i=1}^{N_{\text{on}}} \sum_{t=1}^{|\tau_i|} \text{CLIP}(r_{i,t}(\theta), \hat{A}_i, \epsilon), \tag{6}$$

To further illustrate the meaning of shaping function $f$, we derive the gradient of off-policy objective,

$$\nabla_\theta \mathcal{J}_{\text{SHAPING-OFF}}(\theta) = \mathbb{E}_{\tau \sim \pi_\phi}[f'(\pi_\theta) \underbrace{\frac{\pi_\theta}{\pi_\phi} \nabla_\theta \log \pi_\theta \cdot \hat{A}_j}_{\text{importance sampling}}.] \tag{7}$$

We write $\pi(\tau_{j,t}|q, \tau_{j,<t})$ as $\pi$ for simplicity. From Eq. 7, we can see $f'(\pi_\theta)$ is a weighting function of the gradient. The vanilla mixed-policy GRPO can be regarded as using a linear shaping function, i.e., $f(\pi) = \pi$, where the original importance sampling ratio $\pi_\theta/\pi_\phi$ is applied.

We decompose the log probability and derive the gradient on each output logit (action):

$$\frac{\partial \mathcal{J}_{\text{SHAPING-OFF}}(\theta)}{\partial M_\theta(\tau'_{j,t})} = \mathbb{E}_{\tau \sim \pi_\phi}\left[f'(\pi_\theta)\, \pi_\theta \left[\mathbb{I}(\tau'_{j,t} = \tau_{j,t}) - \pi_\theta\right] \cdot \hat{A}_j\right]$$

$$\Rightarrow \left|\frac{\partial \mathcal{J}_{\text{SHAPING-OFF}}(\theta)}{\partial M_\theta(\tau'_{j,t})}\right| \le \mathbb{E}_{\tau \sim \pi_\phi}\left[|f'(\pi_\theta)|\, \pi_\theta\,(1 - \pi_\theta) \cdot |\hat{A}_j|\right], \tag{8}$$

where $\tau'_{j,t}$ is any possible action/token on in the action space at the $j$-th trajectory and $t$-th position, and $M_\theta(\tau)$ denotes the logits of that action. The identity case represents the gradient when the action is the off-policy action $\tau = \tau'$, which elevates the probability of predicting the off-policy action. From Eq. 8, if $f(\pi_\theta) = \pi_\theta$ and thus $f'(\pi_\theta) = 1$, we can see that the scale of gradient is upper-bounded by $\pi_\theta(1 - \pi_\theta)$, leading to small values when $\pi_\theta \to 0$ and $\pi_\theta \to 1$. We opt for $f(x) = x/(x + \gamma)$ as our shaping function (middle part of Figure 2), where $\gamma$ is set as 0.1. As shown in Figure 2 (Right), the shaping function reweights the gradients to assign more importance to low-probability actions, thereby improving learning from unfamiliar yet effective decisions from the off-policy traces.

Further, we provide an informal analysis on the variance of our importance weights regularized by $f(\cdot)$ in Appendix B.2. With a first-order approximation and a special case derivation, we show that a smaller variance can be achieved for the sampling weights, thus proving more stable training for leveraging off-policy guidance.

## 4 Experimental Setup

**Dataset Construction.** Our training set is a subset of OpenR1-Math-220k [28] [3], of which the prompts are collected from NuminaMath 1.5 [29], and the off-policy reasoning traces are generated by Deepseek-R1 [2]. We use the default subset, which contains 94k prompts, and we filter out generations that are longer than 8192 tokens and those that are verified wrong by *Math-verify*[4], resulting in 45k prompts and off-policy reasoning traces.

**RL Practice.** We remove the KL loss term by setting $\beta = 0$ and set the entropy loss coefficient to 0.01. Following Dr.GRPO[6], we remove the length normalization and standard error normalization of GRPO loss (Eq. 3) for all experiments. For policy shaping, we empirically set the $\gamma$ as 0.1 and study the value of $\gamma$ in Appendix E.4. Our rollout batch size is 128, and the update batch size is 64. We use 8 rollouts per prompt. Specifically, for on-policy RL, we use 8 on-policy rollouts. For our methods, we use 1 off-policy and 7 on-policy rollouts to ensure fairness. We use temperature=1.0 for rollout generation. We use Math-Verify as our reward function and include no format or length reward. We use Qwen2.5-Math-7B [30] by default, following previous work [24, 5, 6]. In addition, we extend LUFFY to Qwen2.5-Math-1.5B [30] and Qwen2.5-Instruct-7B [31], and LLaMA 3.1-8B [32].

**Evaluation.** For evaluation, we mainly focus on six widely used math reasoning benchmarks, including AIME 2024, AIME 2025, AMC [13], Minerva [15], OlympiadBench [14], and MATH-500 [16]. For AIME 2024, AIME 2025 and AMC, we report avg@32 as the test set is relatively small, and for the other three benchmarks, we report pass@1. As our RL training mainly focuses on math reasoning, we further validate the generalization capability on three out-of-distribution benchmarks, namely ARC-c [33](Open-Domain Reasoning), GPQA-diamond [34](Science Graduate Knowledge, denoted as GPQA*), and MMLU-Pro [35](Reasoning-focused Questions from Academic Exams and Textbooks). We shuffle the multiple-choice options to avoid contamination. For testing, the temperature is set as 0.6.

**Baseline Methods.** For RLVR methods, we consider the following methods: (1) *Simple-RL [5]:* training from Qwen2.5-Math-7B using rule-based reward; (2) *Oat-Zero [6]:* training from Qwen2.5-Math-7B and rule-based reward, proposing to remove the standard deviation in GRPO advantage computation and token-level normalization in policy loss computation; (3) *PRIME-Zero [24]*: using policy rollouts and outcome labels through implict process rewards; (4) *OpenReasonerZero [7]:* a recent open-source implementation of RLVR methods. Except RLVR approaches from previous work, we consider two kinds of baselines with our setting (1) *On-Policy RL* – we train on-policy RL within RLVR paradigm using Dr.GRPO with the same reward and data. (2) *Alternative Methods to Incorporate Off-Policy Guidance* – We consider three methods, namely `SFT`, we train the model with the same prompts and reasoning traces as LUFFY using SFT; `RL w/ SFT Loss`, using SFT loss during RL training; `SFT + RL`, two-stage training that continues RL training after SFT. For detailed setup for training these methods, we refer readers to Appendix C.

## 5 Experimental Results

### 5.1 Main Results

**SOTA performance on RLVR with Qwen2.5-Math-7B.** Our main results are presented in Table 1. We first compare LUFFY against other RLVR methods and our RLVR replication. All prior methods are built upon Qwen2.5-Math-7B base models, differing in dataset composition (source and difficulty) and optimization strategies, e.g., removing length and standard error normalization [6] or incorporating process-level rewards [24]. Evaluated on six challenging competition-level benchmarks, LUFFY achieves an average score of **50.1**, significantly outperforming existing RLVR methods by a substantial margin of **+6.4** points, establishing a new state-of-the-art. Notably, while LUFFY exhibits comparable performance in AIME 24, it demonstrates a significantly greater advantage on the newly

---

[3] `https://huggingface.co/datasets/open-r1/OpenR1-Math-220k`
[4] https://github.com/huggingface/Math-Verify

Table 1: Overall in-distribution and out-of-distribution performance based on **Qwen2.5-Math-7B**. We compare with the following baselines: (1) Qwen2.5-Math-7B-Instruct (Qwen-Instruct), (2) prior RLVR approaches, (3) our replication of on-policy RL, and (4) alternative off-policy learning methods. All models are evaluated under a unified setting. LUFFY† denotes training with extra steps (Table 2). Bold and underline indicate the best and second-best results, respectively. ∗ represents significantly better than baselines ($p < 0.05$).

| Model | In-Distribution Performance | | | | | | Out-of-Distribution Performance | | | |
|---|---|---|---|---|---|---|---|---|---|---|
| | AIME 24/25 | AMC | MATH-500 | Minerva | Olympiad | Avg. | ARC-c | GPQA∗ | MMLU-Pro | Avg. |
| Qwen-Base [30] | 11.5/4.9 | 31.3 | 43.6 | 7.4 | 15.6 | 19.0 | 18.2 | 11.1 | 16.9 | 15.4 |
| Qwen-Instruct [30] | 12.5/10.2 | 48.5 | 80.4 | 32.7 | 41.0 | 37.6 | 70.3 | 24.7 | 34.1 | 43.0 |
| *Previous RLVR methods* | | | | | | | | | | |
| SimpleRL-Zero [5] | 27.0/6.8 | 54.9 | 76.0 | 25.0 | 34.7 | 37.4 | 30.2 | 23.2 | 34.5 | 29.3 |
| OpenReasoner-Zero [7] | 16.5/15.0 | 52.1 | 82.4 | 33.1 | 47.1 | 41.0 | 66.2 | 29.8 | **58.7** | 51.6 |
| PRIME-Zero [24] | 17.0/12.8 | 54.0 | 81.4 | 39.0 | 40.3 | 40.7 | 73.3 | 18.2 | 32.7 | 41.4 |
| Oat-Zero [6] | **33.4**/11.9 | 61.2 | 78.0 | 34.6 | 43.4 | 43.7 | 70.1 | 23.7 | 41.7 | 45.2 |
| *Our On-policy RLVR Replication* | | | | | | | | | | |
| On-Policy RL | 25.1/15.3 | 62.0 | 84.4 | 39.3 | 46.8 | 45.5 | **82.3** | 40.4 | 49.3 | 57.3 |
| *Alternative Off-policy Learning Methods* | | | | | | | | | | |
| SFT | 22.2/22.3 | 52.8 | 82.6 | 40.8 | 43.7 | 44.1 | 75.2 | 24.7 | 42.7 | 47.5 |
| RL w/ SFT Loss | 19.5/16.4 | 49.7 | 80.4 | 34.9 | 39.4 | 40.1 | 71.2 | 23.7 | 43.2 | 46.0 |
| SFT+RL | 25.8/**23.1** | 62.7 | 87.2 | 39.7 | 50.4 | 48.2 | 72.4 | 24.2 | 37.7 | 44.8 |
| **Our Methods** | | | | | | | | | | |
| LUFFY | 29.4/**23.1** | 65.6 | **87.6** | 37.5 | **57.2** | 50.1* | 80.5 | 39.9 | 53.0 | 57.8* |
| LUFFY† | 30.7/22.5 | **66.2** | 86.8 | **41.2** | 55.3 | **50.4*** | 81.8 | **49.0** | 54.7 | **61.8*** |

released AIME 25 test set (+8.1), demonstrating its generalization to internalize nuanced reasoning behaviors from off-policy traces. Compared to `On-Policy RL`, LUFFY improves performance by +4.6 points on average, demonstrating the benefit of integrating high-quality off-policy traces.

Regarding out-of-distribution performance, LUFFY also demonstrates strong performance gain. Over three challenging out-of-distribution benchmarks, LUFFY achieves an average score of **57.8** and outperforms the best RLVR method `OpenReasoner-Zero` for **+6.2** points. These findings underscore the effectiveness of LUFFY in leveraging off-policy reasoning guidance for enhanced generalization across diverse, out-of-distribution tasks.

**Comparing against other Off-Policy Learning Methods.** Comparing LUFFY against alternative methods to incorporate Off-Policy Guidance, we can see that LUFFY beats all three off-policy baselines in in-distribution math reasoning tasks and achieves substantial improvements over OOD tasks (**+10.3 points**). Compared to `SFT+RL`, LUFFY is advantageous in both in-distribution tasks (+1.9 points) and out-of-distribution tasks (+16.1), with only 59% GPU hours and much less off-policy data usage (Table 2). The additional GPU hours in `SFT+RL` and `RL w/ SFT Loss` are largely attributed to excessively long generations induced by rigid imitation from SFT (Appendix F), which substantially increase the computational overhead during the RL roll-out stage. With matching GPU hours, LUFFY† further enlarges the advantage, providing a more robust and effective alternative for *distilling knowledge* from stronger LRMs, except for supervised fine-tuning [2, 36, 37]. In

Table 2: **Comparison of resource requirements between LUFFY and other off-policy methods.**

| Model | GPU Hours | Data Usage (On/Off) |
|---|---|---|
| **LUFFY** | $77 \times 8$ | 64K × 7 / 64K |
| **LUFFY†** | $130 \times 8$ | 110K × 7 / 110K |
| **SFT** | $24 \times 8$ | 0 / 64K |
| **RL w/ SFT Loss** | $133 \times 8$ | 64K × 7 / 64K |
| **SFT+RL** | $130 \times 8$ | 64K × 8 / 135K |

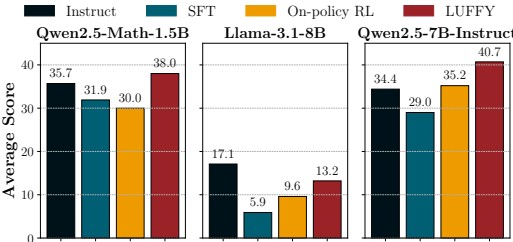

Figure 3: **Average performance on six mathematical reasoning benchmarks of LUFFY on different backbones (Details in Appendix E.2)**

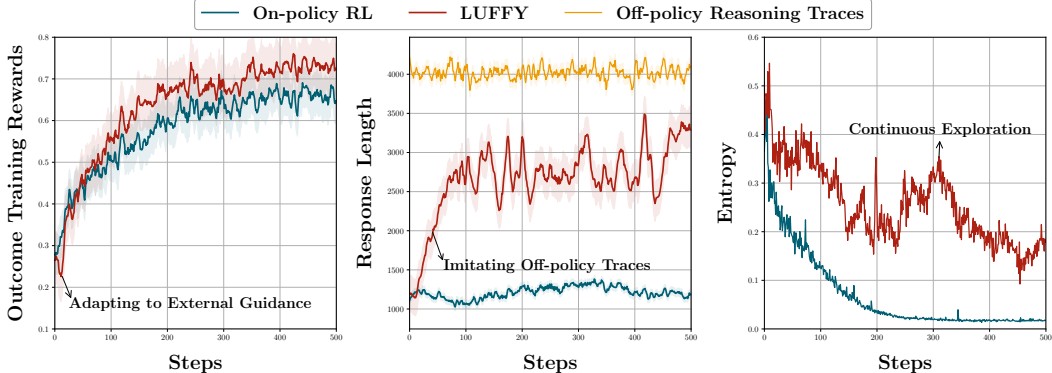

Figure 5: Training dynamics of LUFFY compared with on-policy RL. **Left**: outcome training rewards; **Middle**: generation length; **Right**: generation entropy.

particular, we notice the SFT training causes the model to learn superficial and rigid imitation of off-policy traces, and costs the model on out-of-distribution performance, while LUFFY *selectively* and *strategically* learn from off-policy traces (Sec. 5.3 and App. F) to enhance its own policy rollouts.

**Extending LUFFY to More Models.**    We further investigate whether LUFFY can be applied to *small models, instruction-tuned models, or weak models*. To answer this question, we train LUFFY on three more models, i.e., Qwen2.5-Math-1.5B (small models), Qwen2.5-Instruct-7B (instruction-tuned models), and LLaMA-3.1-8B (weak models), and compare with their respective Instruct models (black bar in Figure 3). LUFFY achieves consistent and substantial improvements, surpassing both `SFT` and `On-Policy RL` for all three models, demonstrating the general applicability of LUFFY. Specifically, LUFFY improves over on-policy RL for +8.0 points on Qwen2.5-Math-1.5B, +3.6 points on Llama-3.1-8B, and +5.5 points on Qwen2.5-Instruct-7B.

## 5.2  LUFFY Succeeds Where On-Policy Fails

More interestingly, we observe that LUFFY can successfully train models in scenarios where on-policy RLVR fails. We conduct experiments using LLaMA-3.1-8B on two subsets of varying difficulty (Easy and Hard), with details provided in Appendix C. As shown in Figure 4, on-policy reinforcement learning performs well on the Easy subset but fails on the Hard subset, where training rewards collapse to zero, since on-policy rollouts struggle to obtain positive feedback signals. In contrast, LUFFY achieves stable reward improvements on both datasets, highlighting its robustness and its ability to *overcome limitations imposed by model capacity*.

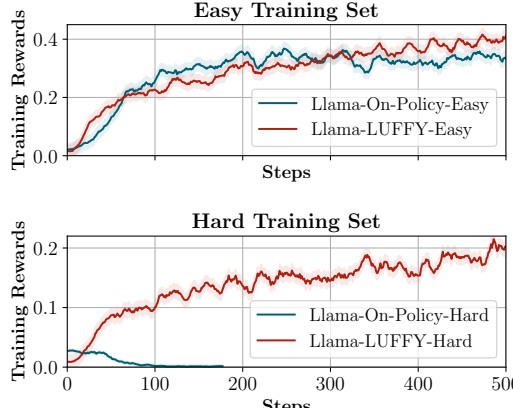

Figure 4: **Training rewards of LLaMA-3.1 8B on the Easy and Hard training set.**

## 5.3  Training Dynamics

**Strategically Learning from Guidance.**    Figure 5 illustrates the training dynamics regarding training rewards, generation length and entropy for `On-Policy RL` and LUFFY. Initially, LUFFY primarily imitates off-policy trajectories, as indicated by the increasing generation length gradually aligning with the off-policy reasoning traces (middle part of Figure 5). At this early stage, imitation dominates, causing an initial performance dip (left part of Figure 5) as the model adjusts to external guidance and potentially sophisticated cognitive behaviors [38]. As training progresses, on-policy rollouts gradually become more prominent, fostering independent exploration within the model's own sampling space while effectively retaining insights gained from off-policy demonstrations. This guided exploration brings growing advantages (training rewards) over `On-Policy RL`. uently, LUFFY achieves a dynamic balance between imitation and exploration, leading to more effective off-policy learning (Section F). These results highlight that LUFFY selectively adopts valuable reasoning patterns

rather than blindly imitating off-policy traces. Such strategic off-policy learning is further evidenced in reasoning behaviors during inference, such as generation length and exploration (Appendix F).

**Maintaining Exploration.**  Figure 5 (Right) illustrates that `LUFFY` consistently sustains higher entropy compared to `On-Policy RL` throughout the entire training process. Specifically, the generation entropy of `On-Policy RL` rapidly converges to nearly zero after approximately 200 steps, indicating a highly deterministic policy with limited exploration potential. Conversely, the elevated entropy observed in `LUFFY` allows continuous exploration of less confident yet potentially superior policies, facilitating the discovery and learning of novel cognitive behaviors. Interestingly, we observe entropy fluctuations and even occasional increases, such as between steps 200 and 250, reflecting ongoing exploration of low-probability but crucial actions, also referred to as *pivotal tokens* [39, 25]. This strategic exploration enables the model to escape local optima, thus improving its convergence towards more globally optimal solutions.

### 5.4  Policy Shaping Encourages Continuous Exploration

Figure 6 illustrates the validation performance over the course of training, shedding light on the impact of policy shaping. `Mixed-Policy` achieves rapid early gains, significantly outperforming `On-Policy RL` at the start. However, its performance soon plateaus and eventually converges with `On-Policy RL`, whereas `LUFFY` continues to improve steadily. These trends are consistent with our earlier analysis of entropy collapse (Section 3.2), underscoring the role of policy shaping as an effective regularizer that prevents premature convergence and sustains performance gains in later stages of training.

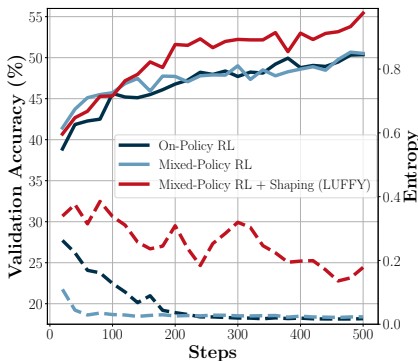

Figure 6: **Effects of policy shaping.**

## 6  Related Work

**RL for LRMs**  Recent advances have demonstrated remarkable progress in enhancing LLMs' reasoning capabilities through RL approaches [2, 1, 3, 40], including DeepSeek-R1, OpenAI-o1, and Kimi-1.5. Subsequent work systematically investigate RL with purely verifiable rewards (RLVR) [5, 7, 24, 6, 41], providing insights into how this approach enables complex reasoning. Various advances have been proposed for reasoning enhancement. Test-time adaptation mechanisms [36, 42] demonstrate potential in dynamic optimization, but remain bounded by inherent model knowledge. While structured reasoning approaches [43, 37, 44, 45] have demonstrated that complex reasoning capabilities can emerge from strategically designed prompting and training strategies. The development of RL optimization techniques [46, 47, 48, 6] has contributed novel training paradigms and optimization objectives that specifically target the enhancement of reasoning capabilities. However, recent studies [8, 9] demonstrate that on-policy learning is limited by vast exploration space and primarily amplifies existing behaviors. Existing approaches optimize within model boundaries rather than expanding reasoning horizons. In the meantime, SFT replaces old capabilities [49] but might not generalize to other domains [50]. Our approach leverages off-policy reasoning traces to transcend these cognitive constraints while preserving self-driven exploration capabilities.

**On-Policy and Off-Policy RL**  Reinforcement learning algorithms are fundamentally distinguished by their approach to experience utilization during policy optimization. On-policy methods (e.g., TRPO [23], A2C/A3C [51], PPO [21]) strictly update using trajectories from the current policy, ensuring training stability but potentially constraining the exploration space. In contrast, off-policy algorithms (e.g., DQN [52],TD3 [53], SAC [54]) leverage experiences from diverse policies, offering superior sample efficiency while introducing optimization challenges due to distribution shift. Extending to LLM training, on-policy methods are more commonly adopted, with approaches like GRPO [12], REINFORCE [55], and PPO [21] demonstrating strong performance through various optimization techniques. PRIME [24] and NFT [56] models the implicit policy to utilize self-generated answers. Meanwhile, off-policy approaches like DPO [57] offer alternative optimization frameworks by reformulating preference learning as classification. To leverage advantages from both paradigms, our work bridges these approaches through policy shaping with regularized importance sampling, effectively combining on-policy optimization with off-policy guidance.

# 7 Conclusion

We presented **LUFFY**, a simple yet powerful framework that integrates off-policy reasoning guidance into the RLVR paradigm. By dynamically balancing imitation and exploration, LUFFY effectively leverages external reasoning traces without sacrificing the model's ability to discover novel solutions. Our method outperforms strong baselines across competitive math benchmarks and generalizes robustly to out-of-distribution tasks, surpassing both on-policy RLVR and off-policy baselines. These results highlight the promise of off-policy learning as a scalable and principled path toward building more general, capable, and self-improving reasoning models. Future work may focus on extending LUFFY to broader domains or modalities [58] and further refining policy shaping to maximize exploration under off-policy guidance.

## Acknowledgement

This publication has emanated from research conducted with the financial support of the National Key RD program of China (grant No.2022YFE0204900) and the National Natural Science Foundation of China Key Program under Grant Number 62336006. This paper focuses on learning to reason from off-policy, which enhances the generalization compared to previous off-policy methods and in-domain reasoning capability compared to on-policy RL.

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

# Appendix

## A  Limitations

Firstly, we mainly focus on math reasoning RL training that has the golden answer and support verifiable rewards. Other tasks, especially ones without verifiable rewards, are not discussed in our manuscript. For instance, in open-ended generation tasks, the reward may be evaluated by an in-domain reward model. Secondly, we focus on 7B and smaller foundation models, due to the limited computational resources. Scaling LUFFY to larger models could be an interesting topic, given scaling law [59, 60] is one of the most powerful principles in the area of large language models. Finally, as we are the first to incorporate off-policy guidance into the RLVR paradigm, we are limited to only including one off-policy trajectory per question, and find that one trajectory is already strong. However, extending off-policy guidance to multiple trajectories and multiple teachers could help the performance even further.

## B  Theoretical Proof

### B.1  Convergence Rate of the Importance-Weighted Policy Gradient Estimator

We study the nonconvex *finite-sum* problems of the form

$$\max_{\boldsymbol{\theta} \in \mathbb{R}^d} J(\boldsymbol{\theta}) := \frac{1}{n} \sum_{i=1}^{n} J_i(\boldsymbol{\theta}), \tag{9}$$

where both $J$ and $J_i$ $(i \in [n])$ may be nonconvex. We denote the class of such finite-sum Lipschitz smooth functions by $J \in \mathcal{J}_n$. Here, we optimize functions in $\mathcal{J}_n$ of our importance-weighted policy gradient estimator.

The vanilla policy gradient algorithm maximizes the expected advantage function (equivalent to minimizing the negative expected advantage function) as

$$\max_{\boldsymbol{\theta} \in \mathbb{R}^d} J(\boldsymbol{\theta}) = \mathbb{E}_{\tau \sim \pi_{\boldsymbol{\theta}}}[A(\tau)] \approx \frac{1}{n} \sum_{i=1}^{n} [A(\tau_i)], \tag{10}$$

According to the Policy Gradient Theorem [61], the vanilla policy gradient estimator has the following form:

$$\nabla J(\boldsymbol{\theta}) = \mathbb{E}_{\tau \sim \pi_{\boldsymbol{\theta}}}[\nabla \log \pi_{\boldsymbol{\theta}}(\tau) \cdot A(\tau)] \approx \frac{1}{n} \sum_{i=1}^{n} [\nabla \log \pi_{\boldsymbol{\theta}}(\tau_i) \cdot A(\tau_i)], \tag{11}$$

where we use $\nabla J(\boldsymbol{\theta})$ to denote $\nabla_{\boldsymbol{\theta}} J(\boldsymbol{\theta})$ for simplicity. Our algorithm draws samples from another behavior policy $\pi_{\boldsymbol{\phi}}$, resulting in an importance-weighted policy gradient estimator as

$$\widetilde{\nabla} J(\boldsymbol{\theta}) = \mathbb{E}_{\tau \sim \pi_{\boldsymbol{\phi}}} \left[ \frac{\pi_{\boldsymbol{\theta}}(\tau_i)}{\pi_{\boldsymbol{\phi}}(\tau_i)} \cdot \nabla \log \pi_{\boldsymbol{\theta}}(\tau) \cdot A(\tau) \right] \approx \frac{1}{n} \sum_{i=1}^{n} [w_i \cdot \nabla J_i(\boldsymbol{\theta})], \tag{12}$$

where $w_i = \frac{\pi_{\boldsymbol{\theta}}(\tau_i)}{\pi_{\boldsymbol{\phi}}(\tau_i)}$ is the importance weight assigned to sample $i$.

Let $\alpha_k$ denote the learning rate at iteration $k$, and $w_{i_k}$ be the instance weight assigned to sample $i$ by our algorithm. By stochastic gradient ascent, our algorithm has the following update rule:

$$\boldsymbol{\theta}^{k+1} = \boldsymbol{\theta}^k + \alpha_k w_{i_k} \nabla J_{i_k}(\boldsymbol{\theta}^k), i \in [n]. \tag{13}$$

**Definition 1.** *For $J \in \mathcal{J}_n$, our algorithm takes an index $i \in [n]$ and a point $x \in \mathbb{R}^d$, and returns the pair $(J_i(\boldsymbol{\theta}), \nabla J_i(\boldsymbol{\theta}))$.*

**Definition 2.** *We say $J : \mathbb{R}^d \to \mathbb{R}$ is Lipschitz smooth (L-smooth) if there is a constant $L$ such that*

$$||\nabla J(\boldsymbol{\vartheta}) - \nabla J(\boldsymbol{\theta})|| \leq L||\boldsymbol{\vartheta} - \boldsymbol{\theta}||, \quad \forall \boldsymbol{\vartheta}, \boldsymbol{\theta} \in \mathbb{R}^d. \tag{14}$$

**Definition 3.** *A point $\boldsymbol{\theta}$ is called $\epsilon$-accurate if $||\nabla J(\boldsymbol{\theta})||^2 \leq \epsilon$. A stochastic iterative algorithm is said to achieve $\epsilon$-accuracy in $k$ iterations if $\mathbb{E}[||\nabla J(\boldsymbol{\theta}^k)||^2] \leq \epsilon$, where the expectation is over the stochasticity of the algorithm.*

**Definition 4.** *We say $J \in \mathcal{J}_n$ has $\sigma$-bounded gradients if $||\nabla J_i(\boldsymbol{\theta})|| \leq \sigma$ for all $i \in [n]$ and $\boldsymbol{\theta} \in \mathbb{R}^d$.*

**Definition 5.** *We say the positive instance weight $w$ in our algorithm is bounded if there exist constants $\underline{w}$ and $\overline{w}$ such that $\underline{w} \leq w_i \leq \overline{w}$ for all $i \in [n]$.*

**Theorem 1.** *Suppose the objective function of the policy gradient algorithm $J \in \mathcal{J}_n$, where $\mathcal{J}_n$ is the class of finite-sum Lipschitz smooth functions, has $\sigma$-bounded gradients, and the importance weight $w = \pi_{\boldsymbol{\theta}}/\pi_{\boldsymbol{\phi}}$ is clipped to be bounded by $[\underline{w}, \overline{w}]$. Let $\alpha_k = \alpha = c/\sqrt{K}$ where $c = \sqrt{\frac{2(J(\boldsymbol{\theta}^*) - J(\boldsymbol{\theta}^0))}{L\sigma^2 \underline{w}\overline{w}}}$, and $\boldsymbol{\theta}^*$ is an optimal solution. Then, the iterates of our algorithm in Eq. (??) satisfy:*

$$\min_{0 \leq k \leq K-1} \mathbb{E}[||\nabla J(\boldsymbol{\theta}^k)||^2] \leq \sqrt{\frac{2(J(\boldsymbol{\theta}^*) - J(\boldsymbol{\theta}^0))L\overline{w}}{K\underline{w}}} \sigma.$$

*Proof.* According to the Lipschitz continuity of $\nabla J$, the iterates of our algorithm satisfy the following bound:

$$\mathbb{E}[J(\boldsymbol{\theta}^{k+1})] \geq \mathbb{E}[J(\boldsymbol{\theta}^k) + \langle \nabla J(\boldsymbol{\theta}^k), \boldsymbol{\theta}^{k+1} - \boldsymbol{\theta}^k \rangle - \frac{L}{2}||\boldsymbol{\theta}^{k+1} - \boldsymbol{\theta}^k||^2]. \tag{15}$$

After substituting (13) into (15), we have:

$$\begin{aligned} \mathbb{E}[J(\boldsymbol{\theta}^{k+1})] &\geq \mathbb{E}[J(\boldsymbol{\theta}^k)] + \alpha_k w_k \mathbb{E}[||\nabla J(\boldsymbol{\theta}^k)||^2] - \frac{L\alpha_k^2 w_k^2}{2} \mathbb{E}[||\nabla J_{i_k}(\boldsymbol{\theta}^k)||^2] \\ &\geq \mathbb{E}[J(\boldsymbol{\theta}^k)] + \alpha_k w_k \mathbb{E}[||\nabla J(\boldsymbol{\theta}^k)||^2] - \frac{L\alpha_k^2 w_k^2}{2} \sigma^2. \end{aligned} \tag{16}$$

The first inequality follows from the unbiasedness of the stochastic gradient $\mathbb{E}_{i_t}[\nabla J_{i_k}(\boldsymbol{\theta}^k)] = \nabla J(\boldsymbol{\theta}^k)$. The second inequality uses the assumption on gradient boundedness in Definition 4. Re-arranging (16) we obtain

$$\mathbb{E}[||\nabla J(\boldsymbol{\theta}^k)||^2] \leq \frac{1}{\alpha_k w_k} \mathbb{E}[J(\boldsymbol{\theta}^{k+1}) - J(\boldsymbol{\theta}^k)] + \frac{L\alpha_k w_k}{2}\sigma^2. \tag{17}$$

Summing (17) from $k = 0$ to $K - 1$ and using that $\alpha_k$ is fixed $\alpha$ we obtain

$$\begin{aligned}
\min_t \mathbb{E}[||\nabla J(\boldsymbol{\theta}^k)||^2] \quad &\leq \frac{1}{K} \sum_{k=0}^{K-1} \mathbb{E}[||\nabla J(\boldsymbol{\theta}^k)||^2] \\
&\leq \frac{1}{K} \sum_{k=0}^{K-1} \frac{1}{\alpha w_k} \mathbb{E}[J(\boldsymbol{\theta}^{k+1}) - f(\boldsymbol{\theta}^k)] + \frac{1}{K} \sum_{k=0}^{K-1} \frac{L\alpha w_k}{2}\sigma^2 \\
&\leq \frac{1}{K\alpha\underline{w}} \left(J(\boldsymbol{\theta}^K) - J(\boldsymbol{\theta}^0)\right) + \frac{L\alpha\overline{w}}{2}\sigma^2 \\
&\leq \frac{1}{K\alpha\underline{w}} \left(J(\boldsymbol{\theta}^*) - J(\boldsymbol{\theta}^0)\right) + \frac{L\alpha\overline{w}}{2}\sigma^2 \\
&\leq \frac{1}{\sqrt{K}} \left(\frac{1}{c\underline{w}}(J(\boldsymbol{\theta}^*) - J(\boldsymbol{\theta}^0)) + \frac{Lc\overline{w}}{2}\sigma^2\right). \tag{18}
\end{aligned}$$

The first step holds because the minimum is less than the average. The second step is obtained from (17). The third step follows from the assumption on instance weight boundedness in Definition 5. The fourth step is obtained from the fact that $J(\boldsymbol{\theta}^*) \geq J(\boldsymbol{\theta}^K)$. The final inequality follows upon using $\alpha = c/\sqrt{K}$. By setting

$$c = \sqrt{\frac{2(J(\boldsymbol{\theta}^0) - J(\boldsymbol{\theta}^*))}{L\sigma^2 \underline{w}\overline{w}}} \tag{19}$$

in the above inequality, we get the desired result. $\qquad\square$

As seen in Theorem 1, our importance-weighted policy gradient estimator has a convergence rate of $O(1/\sqrt{K})$. Equivalently, the time complexity of our algorithm to obtain an $\epsilon$-accurate solution is $O(1/\epsilon^2)$. Note that our choice of step size $\alpha$ requires knowing the total number of iterations $K$ in advance. A more practical approach is to use a time-decayed step size of $\alpha_k \propto 1/\sqrt{k}$ or $\alpha_k \propto 1/k$.

## B.2 Informal Analysis on Variance of Regularized Importance Sampling

Importance sampling is a widely spread Monte-Carlo technique that adopts a reweighting strategy to estimate the so-called target distribution using samples from another distribution. A major drawback of vanilla importance sampling is the large variance of the weights, which is known to impact the accuracy of the estimates badly. In Sec. 3.2, we regularize the importance weights to enhance learning from low-probability tokens with the shaping function:

$$f(x) = \frac{x}{x + \gamma}, \quad \gamma \in [0, 1], \tag{20}$$

where $x = \frac{\pi_{\boldsymbol{\theta}}}{\pi_{\boldsymbol{\phi}}} \in (0, +\infty)$ is the original weight. We consider the first-order approximation of $f(x)$ by Taylor expansion as

$$\begin{aligned}
f(x) &= f(u) + f'(u)(x - u) + \sum_{n=2}^{\infty} \frac{f^{(n)}(u)}{n!}(x - u)^n \\
&\approx f(u) + f'(u)(x - u)
\end{aligned} \tag{21}$$

Suppose that $\pi_{\boldsymbol{\phi}}$ dominates $\pi_{\boldsymbol{\theta}}$, and we have $\mathbb{E}[x] = \mathbb{E}[\frac{\pi_{\boldsymbol{\theta}}}{\pi_{\boldsymbol{\phi}}}] = 1$. We consider the Taylor expansion at point $u = 1$ as

$$f(x) \approx f(1) + f'(1)(x - 1) = \frac{1}{1 + \gamma} + \frac{\gamma}{(1 + \gamma)^2}(x - 1) \tag{22}$$

The variance of the first-order approximation of $f(x)$ is

$$\mathbb{V}ar[f(x)] \approx \mathbb{V}ar\left[\frac{\gamma}{(1+\gamma)^2}(x-1)\right] = \left(\frac{\gamma}{(1+\gamma)^2}\right)^2 \mathbb{V}ar[x] \tag{23}$$

Since $\left(\frac{\gamma}{(1+\gamma)^2}\right)^2 \ll 1$, we have $\mathbb{V}ar[f(x)] \ll \mathbb{V}ar[x]$.

Further, we analyze the variance of the regularized importance weights $f(x) = \frac{x}{x+\gamma}, \gamma \in (0,1)$ given a special case that the original weight variable $x$ follow a specific distribution as $p(x) = e^{-x}, \ x > 0$. This distribution makes sense for the importance weight $x = \frac{\pi_\theta(\tau)}{\pi_\phi(\tau)}$ in our setting. First, with the distribution $p(x) = e^{-x}$, the expectation of $x$ is $\mathbb{E}_{p(x)}[x] = \int_0^\infty e^{-x}\mathrm{d}x = 1$. This matches the expectation of the importance weight as $\mathbb{E}[x] = \mathbb{E}\left[\frac{\pi_\theta(\tau)}{\pi_\phi(\tau)}\right] = 1$, given that $\pi_\phi(\tau)$ dominates $\pi_\theta(\tau)$. Second, the probability density $p(x)$ decreases as $x$ gets larger, which matches the intuition that the importance weight $x = \frac{\pi_\theta(\tau)}{\pi_\phi(\tau)}$ tends to have smaller values. $\pi_\phi(\tau)$ is the probability of an expert trajectory under the assumed optimal policy, which is likely to produce large values as $\pi_\phi(\tau) \to 1$. $\pi_\theta(\tau)$ is the probability of an expert trajectory under the current policy, which is usually small, especially at the early learning stage.

In this case, the variance of the original importance weight is

$$\begin{aligned}
\mathbb{V}ar[x] &= \mathbb{E}_{p(x)}[x^2] - \mathbb{E}_{p(x)}[x]^2 \\
&= \int_0^\infty x^2 e^{-x}\mathrm{d}x - 1 \\
&= 1.
\end{aligned} \tag{24}$$

The variance of the regularized importance weight is

$$\begin{aligned}
\mathbb{V}ar[f(x)] &= \mathbb{E}_{p(x)}[f(x)^2] - \mathbb{E}_{p(x)}[f(x)]^2 \\
&= \int_0^\infty e^{-x}\left(\frac{x}{x+\gamma}\right)^2 \mathrm{d}x - \left(\int_0^\infty \frac{xe^{-x}}{x+\gamma}\mathrm{d}x\right)^2 \\
&= \left(2 - 2\gamma e^\gamma E_1(\gamma) - 2\gamma^2 e^\gamma E_3(\gamma)\right) - (1 - \gamma e^\gamma E_1(\gamma))^2 \\
&= 1 - 2\gamma^2 e^\gamma E_3(\gamma) - \gamma^2 e^{2\gamma} E_1(\gamma)^2 \\
&< 1
\end{aligned} \tag{25}$$

where $E_1(\gamma) = \int_\gamma^\infty \frac{e^{-u}}{u}\mathrm{d}u > 0$ and $E_3(\gamma) = \int_\gamma^\infty \frac{e^{-u}}{u^3}\mathrm{d}u > 0$. Hence, we have $\mathbb{V}ar[f(x)] < \mathbb{V}ar[x]$.

In summary, with the above informal analysis, we show that our regularized importance weights can achieve reduced variance, thus providing more stable training for leveraging off-policy guidance.

## C Experimental Details

### C.1 Detailed Setup

**Easy and Hard Training Set** These two datasets of different difficulties are generated from subsets of the OpenR1-MATH-220K dataset. We first filter the questions for which DeepSeek-R1 can generate a correct answer. Then, we split the data according to the length of DeepSeek-R1's solution. We coin questions R1 can solve within 2k tokens as Easy set and those within 4k tokens as the Hard set, respectively. Intuitively, the more tokens needed for Deepseek-R1 to generate a correct answer, the more difficult the question is. Finally, the Easy dataset contains 7.3k prompts, and the Hard dataset contains 25.4k prompts.

**Training.** In addition to Qwen2.5-Math-7B, we extend LUFFY to Qwen2.5-Math-1.5B [30] and Qwen2.5-Instruct-7B [31], and LLaMA 3.1-8B [32]. To ensure fairness, we maintain 8 samples per prompt for all RL-trained models. The learning rate is constantly set as 1e-6. All training experiments are conducted using 8 A100 GPUs. We train 500 steps for all RL models and three epochs for SFT models. The only exception is LUFFY†, which is trained for 860 steps to match the GPU hour of SFT + RL.

Our implementation is based on verl[5], which uses vLLM[6] as the rollout generators. We are thankful for these open-source repositories.

**Qwen2.5-Series Models.** Since the context length of Qwen2.5-Math is 4096 and the generation length of off-policy samples could be lengthy, we change the rope theta from 10000 to 40000 and extend the window size to 16384. For Qwen2.5-Instruct, the context window is large enough. Hence, we do not change the model configurations. For all Qwen2.5-Series models, we use the same dataset as described in Sec. 4.

**Llama-3.1-8B.** For Llama3.1-8B, we follow Simple-RL-Zoo [20] and use a simplified prompt, and we do not ask the model to generate <think>/</think> tokens.

The dataset used for LLaMA3.1-8B is the subset of OpenR1-Math-220k we used with Qwen2.5-Series models, selected by the length of DeepSeek-R1's correct solution, i.e., 0-2k tokens (Easy training set described in Sec. 5.2). We find that on-policy RL fails on other subsets, such as the Hard training set (0-4k) or the same data used in Qwen2.5-Series.

**SFT.** For all SFT models, we train on the same DeepSeek-R1 generated traces and prompts as LUFFY. We follow the SFT setting from `open-r1/OpenR1-Qwen-7B`[7], which reproduces the performance of Deepseek-R1's distilled model. We train each model for 3 epochs. The train batch size is 64, and the learning rate is 5e-5. We use learning rate warmup ratio 0.1 and set the max length to 16k.

**RL w/ SFT Loss.** For multi-tasking RL and SFT objectives, we compute the on-policy loss on 7 on-policy samples and SFT loss on 1 off-policy sample per prompt. The other setup is the same as other RL experiments.

**SFT + RL** We use the same SFT model described earlier and further conduct RLVR training for 500 more steps. Following previous literature [24], we use the held-out dataset of OpenR1-Math-220k, resulting in around 49k prompts.

## C.2  System Prompt

All our trained models, except LLaMA-3.1-8B, share the same system prompt for training and inference:

> Your task is to follow a systematic, thorough reasoning process before providing the final solution. This involves analyzing, summarizing, exploring, reassessing, and refining your thought process through multiple iterations. Structure your response into two sections: Thought and Solution. In the Thought section, present your reasoning using the format: "`<think>`\n thoughts `</think>`\n". Each thought should include detailed analysis, brainstorming, verification, and refinement of ideas. After "`</think>`\n" in the Solution section, provide the final, logical, and accurate answer, clearly derived from the exploration in the Thought section. If applicable, include the answer in \boxed{} for closed-form results like multiple choices or mathematical solutions.
> **User:** This is the problem: {QUESTION}
> **Assistant:** <think>

For LLaMA-3.1-8B, we do not use the above system prompt as we find the model cannot follow such an instruction. Thus, we use a simplified version that only includes the CoT prompt and do not include `<think>` token.

> **User:** {QUESTION}
> **Answer:** Let's think step by step.

---

[5]https://github.com/volcengine/verl
[6]https://github.com/vllm-project/vllm
[7]https://huggingface.co/open-r1/OpenR1-Qwen-7B

## C.3 Significance Test

We report the significance test results in our main results, i.e., Table 1. The significance test [62] is calculated by paired bootstrapping resampling, and the sample size is 1000 times. The null hypothesis asserts that any observed difference is merely due to random sampling variation rather than representing a genuine effect or difference between the two groups.

From our results, we can see that LUFFY and LUFFY† significantly outperform all baseline methods.

## D  Case Study

A demonstrative case study (Fig.7) comparing our proposed approach (LUFFY) against baseline methods (SFT and GPRO) in mathematical problem solving reveals distinct characteristics in reasoning patterns. SFT demonstrates redundant and circular reasoning with excessive repetition (over 8,129 tokens), while GPRO shows concise but unfounded deduction (1002 tokens), both leading to incorrect conclusions. In contrast, LUFFY presents a well-balanced approach (2623 tokens) that combines systematic decomposition with clear mathematical calculation. Through rigorous reasoning and proper verification steps, LUFFY successfully reaches the correct answer, demonstrating the effectiveness of our methodology in achieving both accuracy and efficiency.

## E  Additional Results

### E.1  Removing On-policy Clip

The clipping mechanism is introduced to constrain policy updates within a trust region [23], thereby ensuring stable training. However, when incorporating off-policy guidance, the target behavior may deviate significantly from the model's current policy, especially early in training.

As shown in Figure 8, `LUFFY` experiences more frequent clipping compared to `On-Policy RL`, which can suppress learning from high-quality off-policy traces. To address this, we remove the on-policy clip to allow greater flexibility in updating toward unfamiliar yet effective actions, thereby unlocking the model's capacity to better integrate off-policy reasoning behaviors.

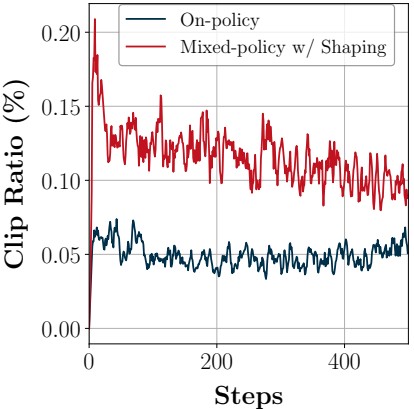

Figure 8: **Ratio of clipped signals.**

### E.2  Extension to More Models

Table 3 presents the detailed performance across six challenging competition-level benchmarks for Qwen2.5-Math-1.5B, Qwen2.5-Instruct-7B, and LLaMA-3.1-8B. On all three models, LUFFY achieves consistent and substantial improvements, surpassing both `SFT` and `On-Policy RL`. On Qwen2.5-Math-1.5B, LUFFY attains an average score of **38.0**, demonstrating notable gains of +6.1 and +8.0 points over `SFT` and `On-Policy RL`, respectively. Similar advantages are observed on the Qwen2.5-Instruct-7B and LLaMA-3.1-8B, where `LUFFY` consistently outperforms baselines across all benchmarks.

### E.3  Ablation Study

In this section, we perform an ablation study to examine the contributions of `LUFFY` components, as summarized in Table 4. Shaping and NoClip both positively contribute to the final performance of Mixed-Policy training. However, applying these enhancements without off-policy guidance

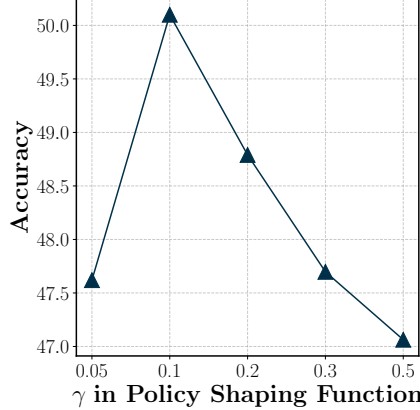

Figure 9: **Accuracy versus the choice of $\gamma$.**

(On-Policy + No Clip/Shaping) does not yield improvement, underscoring the necessity of external signals to acquire nuanced and generalizable reasoning skills.

Table 3: Overall performance on six competition-level benchmark performance on Qwen2.5-Math-1.5B, Qwen2.5-Instruct-7B and LLaMA-3.1-8B.

| Model | AIME 24 | AIME 25 | AMC | MATH-500 | Minerva | Olympiad | Avg. |
|---|---|---|---|---|---|---|---|
| *Qwen2.5-Math-1.5B* | | | | | | | |
| Qwen2.5-Math-1.5B-Base [30] | 7.2 | 3.6 | 26.4 | 28.0 | 9.6 | 21.2 | 16.0 |
| Qwen2.5-Math-1.5B-Instruct [30] | 12.1 | 8.9 | 48.1 | 77.4 | 28.7 | 39.1 | 35.7 |
| SFT | 11.7 | **13.2** | 37.8 | 70.6 | 26.8 | 31.3 | 31.9 |
| On-Policy RL | 11.8 | 7.7 | 40.2 | 61.8 | 26.8 | 32.0 | 30.0 |
| LUFFY | **16.0** | 13.1 | **47.1** | **80.2** | **30.5** | **41.0** | **38.0** |
| *Qwen2.5-Instruct-7B* | | | | | | | |
| Qwen2.5-7B-Instruct [31] | 11.7 | 7.5 | 43.8 | 71.8 | 30.9 | 40.4 | 34.4 |
| SFT | 7.9 | 9.2 | 36.0 | 68.6 | 21.3 | 31.1 | 29.0 |
| On-Policy RL | 14.1 | 8.3 | 43.5 | 74.0 | **33.8** | 37.6 | 35.2 |
| LUFFY | **17.7** | **14.8** | **50.9** | **82.0** | 31.3 | **47.4** | **40.7** |
| *LLaMA-3.1-8B* | | | | | | | |
| LLaMA-3.1-8B-Instruct [32] | 5.1 | 0.4 | 18.6 | 44.6 | 19.5 | 14.1 | 17.1 |
| SFT | 0.5 | 0.1 | 5.4 | 20.2 | 4.0 | 5.3 | 5.9 |
| On-Policy RL | 0.3 | **0.5** | 9.4 | 23.4 | **17.6** | 6.1 | 9.6 |
| LUFFY | **1.9** | 0.1 | **13.5** | **39.0** | 15.1 | **9.6** | **13.2** |

Table 4: Ablation study on LUFFY components.

| Model | AIME 24 | AIME 25 | AMC | MATH-500 | Minerva | Olympiad | Avg. |
|---|---|---|---|---|---|---|---|
| Mixed-Policy RL | 19.4 | 17.7 | 58.9 | 84.6 | 35.7 | 49.9 | 44.4 |
| + Shaping | 27.4 | 21.7 | 61.2 | 86.6 | 37.1 | 53.0 | 47.8 |
| + Shaping + NoClip | 29.4 | 23.1 | 65.6 | 87.6 | 37.5 | 57.2 | 50.1 |
| On-Policy RL | 25.1 | 15.3 | 62.0 | 84.4 | 39.3 | 46.8 | 45.5 |
| + Shaping | 21.3 | 13.6 | 58.0 | 80.6 | 36.8 | 41.8 | 42.0 |
| + No Clip | 21.5 | 17.4 | 61.1 | 83.4 | 36.8 | 49.0 | 44.9 |

## E.4 Hyperparameter Study

In this section, we study the choice of $\gamma$ in policy shaping function. The results are shown in Figure 9, trained from Qwen2.5-Math-7B. We choose $\gamma$ value from [0.05, 0.1, 0.2, 0.3, 0.5]. When $\gamma = 0.1$, the model performs the best with 50.1 accuracy scores, and increasing or decreasing this value leads to a notable decline in model performance. Therefore, we consistently use $\gamma = 0.1$ throughout our experiments.

## F Analysis

In this section, we analyze how LUFFY effectively leverages off-policy guidance, i.e., *imitating to illuminate*, to improve reasoning quality and generalization.

### F.1 LUFFY Learns Strategically from Off-Policy Traces, While SFT Imitates Rigidly

We compare the generation length distributions of LUFFY and SFT on the combined set of six mathematical reasoning benchmarks. As shown in Figure 10, LUFFY produces significantly shorter generations on average (2,832 tokens) compared to SFT (4,646 tokens), suggesting a more effective reasoning process that balances imitation and exploration. This observation helps explain the

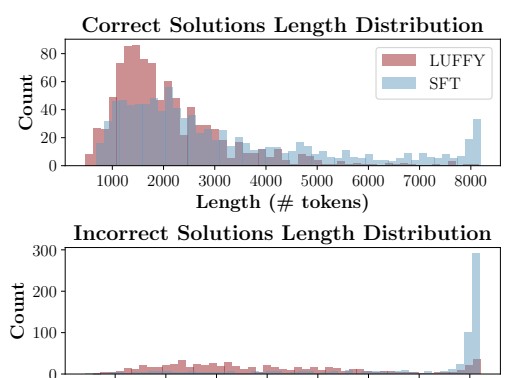

Figure 10: **Generation length of correct and incorrect solutions.**

excessive training costs of methods that naively combine SFT and RL, e.g., SFT+RL and RL w/ SFT Loss (Table 2), as these models spend substantially more compute on producing unnecessarily lengthy CoTs during the RL roll-out stage. In contrast, SFT often mimics the surface form of off-policy demonstrations without genuinely engaging in problem-solving. This behavior is especially evident in incorrect outputs, where SFT frequently generates overly long and ultimately unproductive reasoning traces. These results indicate that while both methods are exposed to similar off-policy signals, LUFFY learns to selectively internalize useful reasoning patterns, whereas SFT tends to overfit to superficial features of the off-policy data.

We further analyze the generation lengths of RL w/ SFT Loss and LUFFY during training, as shown in Figure 11. RL w/ SFT Loss quickly imitates the off-policy traces, exhibiting a steep increase in generation length early in training. However, it soon becomes trapped in the superficial patterns of the demonstrations, leading to excessively long outputs that even surpass the length of the original off-policy traces. In contrast, LUFFY's dynamic advantage balancing between on-policy and off-policy rollouts encourages more strategic learning. As a result, its generation length grows more gradually and steadily, reflecting a more selective and grounded adoption of reasoning behavior.

Beyond generation length, imitation behavior can also be observed through the similarity between model outputs and off-policy traces. To quantify this, we compare generations from SFT, On-Policy RL, and LUFFY against those from DeepSeek-R1 on a held-out set of 1,000 samples, using BLEU [63] as the similarity metric. The resulting

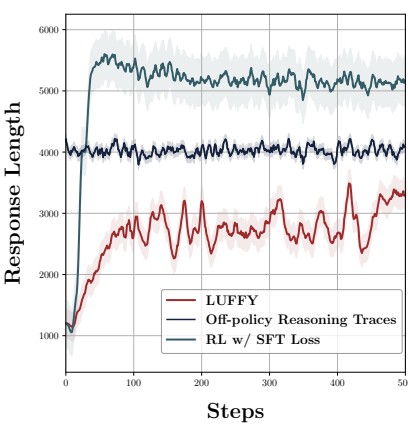

Figure 11: Generation length of RL w/ SFT Loss and LUFFY.

BLEU scores are 57.5 for SFT, 8.8 for On-Policy RL, and 44.8 for LUFFY, reflecting the strong imitation behavior of SFT and the more selective, yet substantial, imitation in LUFFY. We present a case study in Appendix D.

## F.2 LUFFY Can Explore During Test-time While SFT Cannot.

We compute pass@8 accuracy on the combined AIME 2024 and AMC datasets, varying the generation temperature from 0.1 to 1.0. As shown in Figure 12, both RL-based methods (On-Policy RL and LUFFY) exhibit strong exploratory capabilities, with pass@8 improving as the temperature increases, showing potential in scaling test-time compute [64]. In contrast, although SFT performs comparably to LUFFY under near-deterministic decoding (temperature 0.1), its performance deteriorates at higher temperatures, failing to uncover additional correct reasoning paths. This highlights the fragility and limited adaptability of SFT, which aligns with prior findings [65, 66] that suggest SFT tends to memorize reasoning patterns rather than learning generalizable reasoning capability.

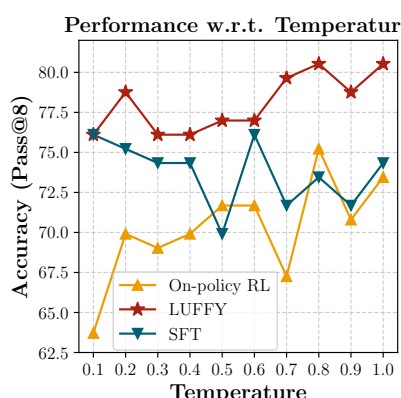

Figure 12: **Pass@8 accuracy (on the merge sets of AIME 2024 and AMC) under different generation temperatures.**

## Case Study

**Prompt:** *\<system prompt\>* This is the problem: A diagonal of a regular 2006-gon is called odd if its endpoints divide the boundary into two parts, each composed of an odd number of sides. Sides are also regarded as odd diagonals. Suppose the 2006-gon has been dissected into triangles by 2003 nonintersecting diagonals. Find the maximum possible number of isosceles triangles with two odd sides.

| SFT | On-Policy RL | LUFFY |
|---|---|---|

**SFT**

**Response:**
*\<think\>*
[Problem Restatement Not Analysis]
Okay, let's try to tackle this problem.
So, we have a regular 2006-gon, and we need to find the maximum number of isosceles triangles...
[Endless Definition Loop]
Let me break this down step by step. First, let's recall some basics about regular polygons and their diagonals...
Hmm, maybe they mean that sides are considered as diagonals here?...
Hmm, maybe they are using "diagonal" in a different sense here...
Alternatively, maybe the problem is using "diagonal" to mean any chord...
[Wait-Pattern Paralysis]
Wait, but in the problem statement...
Wait, but sides are chords...
Wait, but in the problem's definition, sides are odd diagonals...
[Small-Case Trap]
Let's take a regular polygon with a small even number of sides, say 4 (a square). A triangulation of a square has 2 triangles...
[Complete Breakdown]
but diagon diagon polygon.< odd diagon in the odd of odd diagon = think... wait diagon diagon diagon diagon...
**Tokens Length:** $> 8192$
**Correctness:** False

**On-Policy RL**

**Response:**
*\<think\>*
[Formulaic Decomposition]
To solve this problem, we need to understand ...
[Pseudo-Systematic Structure]
Let's step-by-step.
1.Understanding the ... if we label the vertices of the polygon as $(V\_1,...,V\_2006)$...
[Circular Property Description]
Properties of Odd Diagonals... Properties of triangulation... Properties of the polygon...
[False Authority Appeal]
The number of isosceles triangles with two odd sides in a triangulation of a regular polygon is a well-known problem in combinatorial geometry...
[Unsubstantiated Conclusion]
For a 2006-gon, the maximum number of such triangles is 501.
*\</think\>*
[Final Answer]
The maximum possible number of isosceles triangles with two odd sides in a triangulation of a regular 2006-gon is $\boxed{501}$.
**Tokens Length:** 1002
**Correctness:** False

**LUFFY**

**Response:**
*\<think\>*
[Clear Problem Definition]
To solve this problem, we need to understand ... and how it can be ... We are specifically interested in ...
[Systematic Decomposition and Analysis]
First, let's define some terms and properties...
Step 1: Understanding Odd Diagonals...
Step 2: Properties of Isosceles Triangles in a Regular Polygon...
Step 3: Counting Odd Diagonals...
Step 4: Dissecting the Polygon into Triangles...
[Precise Mathematical Calculation]
"For each vertex, there are 1002 vertices at an odd distance...
[Logical Verification]
Given that each isosceles triangle with two odd sides uses one odd diagonal and one side...
[Consistency Check]
"Let's verify the logic:
1. Odd Diagonals ...
2. Dissection ..."
*\</think\>*
[Final Answer]
Thus, the maximum possible number of isosceles triangles with two odd sides is $\boxed{1003}$.
**Tokens Length:** 2623
**Correctness:** **True**

**Answer:** "$1003$"

Figure 7: Comparison of three approaches (SFT, On-Policy RL, and LUFFY) for a geometric problem.

