# OpenReview forum: "Learning to Reason under Off-Policy Guidance"
_NeurIPS.cc/2025/Conference — NeurIPS 2025 poster_

### Official Review · Reviewer_ubx4 · 2025-07-02

**Clarity:** 3
**Significance:** 2
**Originality:** 2
**Rating:** 4
**Confidence:** 4

**Summary:**

This paper focuses on off-policy RL for LLMs. Specifically, the proposed method mixes in some off-policy rollouts during RL training (e.g., 7:1 on-policy to off-policy in some setups), and applies a shaping function x / (x + \gamma) to the importance sampling ratio in the policy objective. They show that with this setup, the model outperforms baselines on a math dataset after RL.

**Questions:**

In Table 2, why do the data setups vary so much across different experiments?

**Ethical Concerns:**

["NO or VERY MINOR ethics concerns only"]

**Final Justification:**

This work directly uses distillation data as off-policy RL rollouts to perform GRPO, and the reported numbers look quite decent. They introduce some tricks, such as transforming the importance sampling ratio to avoid strictly adhering to imitation-labeled data.

The method is very straightforward, with limited novelty.

A key concern is that, given the maturity of existing SFT+RL infrastructures, a more carefully designed pipeline may perform just as well or even better than this trick-based approach.

The authors argue that their two-stage method saves GPU hours, but their experiments do not align the consumed data across different settings, which calls for more controlled experiments to validate the claim. I remain cautious about this point.

Therefore, I give a weak positive score.

**Limitations:**

yes

**Quality:**

2

**Strengths And Weaknesses:**

### Pros:
- Clear and easy-to-follow writing
- Shows positive results empirically

### Cons:
- The method is pretty straightforward—more like a numerical trick. The actual effectiveness and necessity need more ablations. For instance, could the “policy shaping” part just be done with a well-tuned clipping strategy?
- The ablation study setup feels messy. The gains are questionable. The method is not purely off-policy; it uses a mix of off- and on-policy data. Most baselines use different rollout sources. In other words, compared to the RLVF baseline, this method uses extra distillation from DeepSeek-R1 (so the performance boost is somewhat expected). Compared to SFT, it also benefits from online rollouts, which could help with OOD robustness. It is not apples-to-apples.
- More metrics like pass@k would help better reflect actual effectiveness, instead of just reporting average reward.

---

> ### Author Rebuttal · Authors · 2025-07-30
>
> We thank you for your time and valuable feedback. We appreciate you highlighting the **"Clear and easy-to-follow writing"** of our paper and for acknowledging that our method **"Shows positive results empirically."** Also, we are grateful that **other reviewers** (we refer to VkPb as R1 and 5T3H as R2) **are quite affirmative to our overall contributions, including the motivation and novelty** (”is very relevant to the contemporary advances…addresses an important issue” by R1 and “ clearly motivate their algorithm design” by R2), **the empirical evaluation** (”the experimental setup is solid…the ablations are complete and comprehensive” by R1 and “provide comprehensive evaluation…a set of comprehensive baselines” by R2), **and the writing** (”the paper is well written” by R2).
>
> Before addressing your question, we first want to offer this concise clarification of our research question and contributions to ensure we are on the same page.
>
> ***Research Question:***
>
> - On-policy Reinforcement Learning from Verifiable feedback (RLVR) is the de facto method for training reasoning models, shown by Deepseek-R1, but its performance is capped by the base model's capability, limiting the usage of RLVR for LLMs[1,2], especially ones without strong initial performance. Our research question is: **How can we use off-policy guidance from a stronger model to break this performance ceiling?** This is a timely and critical issue for advancing LLM reasoning, as recognized by Reviewer VkPb.
>
> ***Our Contributions:***
>
> 1. **A Novel and Effective Mixed-Policy Method for RLVR.**
>     - Our method, LUFFY, is the first to use a mixed-policy framework to **meaningfully improve** RLVR performance, not just accelerate it.
>     - This is a non-trivial contribution. We show that naive mixed-policy fails to improve final results (as discussed in Section 3.2 and especially L156-L161). Our key insight is to introduce **policy shaping** to solve the *entropy collapse* problem, forcing the model to learn from informative but low-probability tokens.
> 2. **Strong Empirical Validation Proving Our Thesis.**
>     - We establish a **new SOTA** on RLVR with Qwen2.5-Math-7B.
>     - We demonstrate **generalizability** across various foundation models.
>     - Crucially, we show LUFFY **successfully trains weaker models where on-policy RLVR fails**, directly proving that our method overcomes the base model's capacity limits (our RQ).
> - We hope this clarifies the significance and novelty of our work.
>
> **Q1: *The method is pretty straightforward—more like a numerical trick. The actual effectiveness and necessity need more ablations. For instance, could the “policy shaping” part just be done with a well-tuned clipping strategy?***
>
> - Our method is far more than a mere numerical trick. Foremost, we propose **a pure RL-based method to address the significant challenge** of empowering LLMs to acquire reasoning behaviors surpassing their initial cognitive boundaries. We provide an elegant way to harness the benefits of both worlds (the external guidance and self-driven exploration) via a unified mixed-policy update rule. **Unlike SFT, we novelly formulate the external guidance as an off-policy objective within a pure RL framework**. It enables the model to internalize deeper and more generalizable reasoning behaviors through the lens of RL. Then, we propose policy shaping via regularized importance sampling to enhance exploration by emphasizing low-probability tokens. Theoretical analyses are provided to highlight the convergence rate of our mix-policy update and the better training stability achieved by our policy shaping.
> - Furthermore, we have the experimental results to justify why policy shaping is both **necessary** and **non-trivial**:
> - *Ablation Study Demonstrates Necessity (Table 4):*
>   - Our ablation study (Table 4) shows that neither mixed-policy GRPO nor policy shaping alone yields substantial improvements. Only when both are combined does the model achieve significant gains. This highlights that policy shaping is critical for encouraging the model to learn from low-probability but crucial decisions. Without shaping, the model tends to overfit high-probability actions from the off-policy traces, leading to entropy collapse.
> - *Shaping vs. Clipping are Fundamentally Different:*
>   - Policy shaping is not a mere numerical trick like clipping. Shaping re-weights gradients to emphasize low-probability yet pivotal tokens, enabling the model to internalize key reasoning steps from off-policy trajectories. In contrast, clipping is a constraint that limits updates for actions deviating too much from the current distribution. Importantly, our method does not use clipping on off-policy rollouts, as explicitly stated in Sec. 3.1.
> - *Clipping Alone is Insufficient (Figure 6):*
>   - We have tried mixed-policy GRPO with different clipping values, and all results were comparable to plain mixed-policy (Figure 6). This confirms that clipping cannot replicate the effect of shaping, which is designed to actively encourage exploration of critical low-probability actions.
>
> ***Q2: The ablation study setup feels messy. The gains are questionable. The method is not purely off-policy; it uses a mix of off- and on-policy data. Most baselines use different rollout sources. In other words, compared to the RLVF baseline, this method uses extra distillation from DeepSeek-R1 (so the performance boost is somewhat expected). Compared to SFT, it also benefits from online rollouts, which could help with OOD robustness. It is not apples-to-apples.***
>
> - We respectfully disagree and would like to clarify:
> - *Ablation Study Highlights Non-Trivial Gains:*
>   - Beyond the main results, our ablation study (Table 4) carefully compares on-policy, mixed-policy, and full LUFFY. This setup demonstrates that mixed-policy alone does not yield the same improvements, and policy shaping is critical to leveraging off-policy data effectively. These controlled ablations underline that LUFFY’s improvements are not from superficial data mixing but from a well-designed algorithmic combination of mixed-policy GRPO and shaping.
> - *Comprehensive and Fair Baseline Coverage:*
>   - With all due respect, we believe our comparisons are thorough and fair, as noted by Reviewer 5T3H and VkPb. Since LUFFY is the first mixed-policy approach in RLVR, we compare against:
>     - Pure on-policy: Standard RLVR baselines and our replication of on-policy RL.
>     - Pure off-policy: SFT using the same DeepSeek-R1 traces.
>     - Mixed-policy: RL w/ SFT Loss and SFT+RL.
>   - These cover all three categories (on-policy, off-policy, mixed-policy). LUFFY achieves substantial gains both on in-domain and out-of-distribution benchmarks compared to these baselines, with comparable data and rollout sources.
> - *Fairness of Data Usage:*
>   - The comparison with SFT and SFT+RL is strictly fair — all methods use the same off-policy traces (from DeepSeek-R1) and online rollouts. LUFFY’s superior OOD robustness stems from its ability to balance exploration and imitation, which SFT and SFT+RL fail to achieve due to their tendency to overfit to fixed off-policy data.
>
> ***Q3: More metrics like pass@k would help better reflect actual effectiveness, instead of just reporting average reward.***
>
> - Thanks for your advice. We have the results of pass@8 in Appendix F.2 due to the space limit. With increased temperature, the pass@8 of On-Policy and LUFFY go up and SFT drops, and LUFFY has a substantial improvement.
> - The results demonstrate that LUFFY can maintain better exploration with large potential solution space while SFT cannot. We will modify our main content to better reference these results. Also, we have conducted experiments with confidence interval to demonstrate the effectiveness of LUFFY is not affected by variance. Please refer to our response to Reviewer VkPb.
>
> ***Q4: In Table 2, why do the data setups vary so much across different experiments?***
>
> - We are sorry for the confusion caused. Here, we aim for a fair comparison in terms of either data or GPU usage, to demonstrate the effectiveness of LUFFY.
> - The data usage represents the data generated and consumed during the RL phase. We split the usage into on-policy usage and off-policy usage to demonstrate how much off-policy data different methods rely on. LUFFY is our base setup and LUFFY+ is the setup we match GPU hours with that of SFT+RL. We will make that part clearer in our next version.
>
> Please let us know if we have addressed your concerns. We are more than delighted to have further discussions and improve our manuscript. **If our response has addressed your concerns, we would be grateful if you could re-evaluate our work.**
>
> *[1] Yue, Y., Chen, Z., Lu, R., Zhao, A., Wang, Z., Song, S., & Huang, G. (2025). Does reinforcement learning really incentivize reasoning capacity in llms beyond the base model?. arXiv preprint arXiv:2504.13837.*
>
> *[2] Rajani, N., Gema, A. P., Goldfarb-Tarrant, S., & Titov, I. (2025). Scalpel vs. Hammer: GRPO Amplifies Existing Capabilities, SFT Replaces Them. arXiv preprint arXiv:2507.10616.*

---

> ### Author Response · Authors · 2025-08-05
>
> Dear Reviewer ubx4,
>
> Thank you again for your valuable time in reviewing our work and for your constructive feedback. We posted our detailed response to your comments approximately five days ago. As the end of the discussion stage is approaching, we were wondering if you might have had a chance to look at it. We would be very keen to hear your thoughts so we can ensure we have fully addressed your concerns.
>
> In our previous response, we
>
> - **Clarified the significance of our research question and contributions.**
>
> - **Clarified that 'policy shaping' is a principled algorithmic solution, not a numerical trick.**
>
> - **Justified the fairness and comprehensiveness of our comparisons.**
>
> We would appreciate it if you could kindly take a look at our full response. If you have any further questions, we are very happy to discuss them!
>
> Best regards,
>
> All authors of Paper 7573

---

> > ### Comment · Reviewer_ubx4 · 2025-08-06
> >
> > I appreciate the authors’ detailed follow-up clarifications. Also, thanks for your explanations regarding comparison. And please incorporate a clearer explanation of the different consumption of data, such as that shown in Table 2, in the revised paper.
> > I do recognize the paper provides a somewhat alternative to the popular distillation + RL training pipeline.
> > However, I still find the evaluation somewhat weak in terms of **controlled** ablation experiments as aligning every consumed rollout, both on-policy and off-policy, and suggest that the authors provide them in future revisions.
> >
> > Overall, the rebuttal addresses some of my concerns, and I will update the rating to 4 to reflect this.

---

### Official Review · Reviewer_5T3H · 2025-07-03

**Clarity:** 3
**Significance:** 3
**Originality:** 3
**Rating:** 5
**Confidence:** 3

**Summary:**

This work presents a new algorithm that blends on-policy GRPO with off-policy policy optimization from expert demonstrations. This algorithm avoids pure on-policy RL training that can fail to learn novel behavior. Hence, the authors propose to use guidance from stronger models (off-policy reasoning traces) to improve performance. They empirically evaluate their method in Math reasoning tasks.

**Questions:**

- From Equation 6 to 7, the function f changes from being a function of the ratio \pi_theta / \pi_\phi to only depend on \pi_\theta. Why is that?
- Why does your on-policy RLVR replication have an overall improvement of performance with respect to the baselines?
- Why is setting \pi_\theta = 1 a reasonable choice?

**Ethical Concerns:**

["NO or VERY MINOR ethics concerns only"]

**Final Justification:**

The author addressed the question I had during the review process. I find the paper to be well motivated and addresses a problem that might be of interest in the community. Moreover, their evaluation show empirical merit to their method.

**Limitations:**

Yes

**Paper Formatting Concerns:**

No major formatting issues

**Quality:**

3

**Strengths And Weaknesses:**

Strengths:
- The paper is well written. The authors clearly motivate their algorithm design.
- The authors provide a set of comprehensive baselines to contextualize the performance of their method.
- The authors provide empirical evidence of the algorithms training dynamics which provide further insight of their method.

The authors provide comprehensive evaluation of their method and introduce a practical shaping mechanism to avoid collapse of the policy’s entropy that their basic mixed-policy GRPO. Moreover, they show empirically that they are able to exploit the expert’s guidance to improve upon the SFT+RL pipeline that is common practice.

---

> ### Author Rebuttal · Authors · 2025-07-30
>
> We would like to extend our sincere thanks for your positive and encouraging review. We are delighted that you found the paper to be "**well written**" and our algorithm design "**clearly motivated**." We particularly appreciate you highlighting our "**comprehensive evaluation**," the inclusion of strong baselines, and our analysis of the training dynamics which you noted provide "**further insight**" into our method. Finally, we are very grateful for your acknowledgment that we were able to empirically show improvement upon the common SFT+RL pipeline. Your thoughtful feedback is incredibly motivating.
>
> The following is our response to your questions:
>
> ***Q1: From Equation 6 to 7, the function f changes from being a function of the ratio \pi_theta / \pi_\phi to only depend on \pi_\theta. Why is that? Why is setting \pi_\theta = 1 a reasonable choice?***
>
> - In our methodology, we simplify the policy ratio from p_theta / p_phi to solely p_theta. This simplification stems from treating p_phi as an optimal reference policy and is a pragmatic choice driven by two key practical considerations.
> - Conceptually, we regard the reference policy p_phi as an oracle or an "expert" that generates the ground-truth or target responses. Under this assumption, the actions (tokens) in the reference sequence are considered the definitive optimal choices. Therefore, the probability of the optimal policy taking these actions is implicitly treated as 1 (i.e., p_phi = 1), simplifying the ratio to just the probability of our own policy, p_theta.
> - This design choice is motivated by the following challenges:
>   - *Tokenization Mismatches:* The expert policy we aim to learn from (e.g., a powerful proprietary LLM) often employs a different tokenizer than our trainable policy p_theta. This tokenization mismatch makes a direct, token-by-token probability comparison infeasible. For instance, the same text could be segmented into token sequences of different lengths by the two models, rendering the computation of a meaningful probability ratio impossible.
>   - *Compatibility with Existing Datasets:* This simplification facilitates the direct use of popular off-the-shelf datasets (e.g., instruction-following or dialogue corpora). These public resources typically provide high-quality text demonstrations but don't include the generation probabilities from the original source model. Our approach bypasses the need to recompute or approximate p_phi—which is often an impractical or impossible task—thereby significantly enhancing the real-world applicability of our method.
> - In L142 to L149 we discuss our reasons and we will modify this part to make it clearer.
>
> ***Q2: Why does your on-policy RLVR replication have an overall improvement of performance with respect to the baselines?***
>
> - The main reason we think is the **better choice of dataset with** **proper difficulty**. SimpleRL-Zero and Oat-Zero use the MATH dataset. OpenReasoner collects both in-domain math problems (e.g., AIME (up to 2023), MATH, Numina-Math collection, etc.) and synthesizes general reasoning tasks using programmatic approaches.  PRIME uses an 860K subset of Numia-Math.
> - In comparison, we use a subset of the OpenR1 dataset by filtering the correctness of Deepseek-R1's trajectories. There is recent work discussing how the difficulty of the dataset affects the successful training and final performance of RLVR[1,2]. Their findings demonstrate that either the dataset is too difficult or too easy for the model would harm the final performance of RLVR.
> - As our on-policy baseline has a better performance, it further validates the effectiveness of LUFFY.
>
> We hope our responses answer your questions, and again, thanks for your advice to improve our paper!
>
> _[1] Chen, M., Liu, H., Liang, H., Huang, H., Zhang, W., & He, R. (2025). Unlocking the Potential of Difficulty Prior in RL-based Multimodal Reasoning. arXiv preprint arXiv:2505.13261._
>
> _[2] Wang, Y., Yang, Q., Zeng, Z., Ren, L., Liu, L., Peng, B., ... & Shen, Y. (2025). Reinforcement learning for reasoning in large language models with one training example. arXiv preprint arXiv:2504.20571._

---

> > ### Comment · Reviewer_5T3H · 2025-08-06
> >
> > Thank you for your responses to my question. I appreciate it!

---

### Official Review · Reviewer_VkPb · 2025-07-05

**Clarity:** 3
**Significance:** 4
**Originality:** 3
**Rating:** 5
**Confidence:** 4

**Summary:**

This paper proposes the idea of introducing policy guidance in the RLVR framework. The idea of using off policy demonstrations have been very effective in RL literature and this paper introduces a similar term in GRPO objective especially for hard datasets where online RL is not able to take off and does not provide any rewards. Introduction of this objective further leads to entropy collapse where the model focuses only on off policy trajectories. This is further addressed by introducing policy shaping in the objective. Experiments with open source Qwen and Llama models show significant performance improvements. The experimental setup looks solid with comparison with many baselines including alternatives to off policy guidance.

**Questions:**

Confidence intervals are missing for all the results. It is difficult to ascertain gains without confidence intervals. How much is variance when sampling results especially performance @32

**Ethical Concerns:**

["NO or VERY MINOR ethics concerns only"]

**Final Justification:**

I have read other reviews and other feedback. I would encourage authors to include the rebuttal results and discussion in the revision and would like to maintain my current score.

**Limitations:**

It would be important to highlight and include discussions on dataset where Luffy didn’t perform that well compared to other methods e.g Arc-C or Minerva

**Paper Formatting Concerns:**

The legend in Figure 6 is not clear. Although it can be understood that dotted lines represent entropy, it should be labelled.
Line 296: typo as “uently”

**Quality:**

3

**Strengths And Weaknesses:**

Strengths:
* This paper proposes the idea of using off policy trajectories with RLVR which is very relevant to the contemporary advances in LLMs for reasoning so it is very timely and addresses an important issue how to bootstrap RL methods when on policy trajectories fetch no rewards
* The experimental setup is solid with comparison across a variety of methods including other methods which can use off policy trajectories including variations of using SFT and experiments on 6 different benchmarks.  The ablations are complete and comprehensive
* The section to compare against other off policy learning methods is really interesting especially from GPU usage point of view and gives an interesting alternative to distillation methods
Weaknesses:
* There is some confusion in the presented theory. Equation 4 mentions mean and std in advantage is computed jointly over both off policy and on policy trajectory. Was the same used in equation 5 (2nd term). How is A_j computed in equation 5 ? Would it be better to clarify this aspect clearly with each term clearly explained for reproducibility ?
* The paper presentation on theoretical parts is unclear although complete and could be improved significantly. For example, the jump from eq 7 to eq 8 could be described in more detail.
* The limitations in datasets where there is performance loss should be discussed and pointed out clearly especially in table 1

---

> ### Author Rebuttal · Authors · 2025-07-30
>
> We would like to sincerely thank you for your detailed and constructive review. We are very encouraged by your positive feedback and appreciate that you recognized our work as "**very timely**" and addressing an "**important issue**." We are especially grateful for your comments highlighting our "**solid**" and "**comprehensive**" experimental setup, the significant performance improvements, and the completeness of the ablation studies. It was also wonderful to hear that you found our comparison to other off-policy methods "**really interesting**" and a valuable alternative. Your encouraging assessment is greatly appreciated, and we will carefully address your suggestions for improvement to further strengthen the paper.
>
>
> The following is our response to your questions:
>
> **Q1: There is some confusion in the presented theory. Equation 4 mentions mean and std in advantage is computed jointly over both off policy and on policy trajectory. Was the same used in equation 5 (2nd term). How is A_j computed in equation 5 ? Would it be better to clarify this aspect clearly with each term clearly explained for reproducibility ?**
>
> - The advantage in equation 5 is the same as equation 4. We use \hat{A}_i and \hat{A}_j to denote that they are from off-policy and on-policy trajectories, respectively. Thanks for your comment, and we will clarify this aspect in our next version.
>
>
>
> **Q2: The jump from eq 7 to eq 8 could be described in more detail.**
> - Thanks for your advice. The derivation from Equation 7 to Equation 8 makes use of the derivative of the softmax function. We will introduce more intuition about how we derive it and put a more detailed derivation in the appendix for reference.
>
> **Q3: It would be important to highlight and include discussions on dataset where Luffy didn’t perform that well compared to other methods e.g Arc-C or Minerva.**
>
> - That’s an insightful question! For Minerva, we manually checked the dataset and find the dataset is sometimes noisy and the golden answer is not perfect. We observe that all methods bring steady improvements on in-domain benchmarks, except for Minerva, which fluctuates a lot.
> - For OOD benchmarks like ARC-C, We do not interpret LUFFY’s performance on ARC-C as a loss. In fact, compared with SFT-based methods (SFT, SFT+RL, RL w/ SFT loss), which show significant drops on ARC-C, LUFFY performs on par with on-policy RL, maintaining strong OOD generalization. This highlights that our mixed-policy approach effectively leverages in-domain knowledge without erasing out-of-domain capabilities—a key advantage over naive off-policy distillation.
> - We will add more discussion about this in our next version to further illustrate this issue.
>
> **Q4: Confidence intervals are missing for all the results. It is difficult to ascertain gains without confidence intervals. How much is variance when sampling results especially performance @32.**
>
> - Thanks for your thoughtful comment. We have repeated our evaluation with AMC, AIME, AIME 25 for 32 runs and other benchmarks with 3 runs to compute the confidence interval. The shown results are the 95% confidence interval and computed by 1.95 * standard deviation. The results with confidence intervals are as follows:
>
> | Model | MATH | Minerva | Olympiad | AIME | AIME25 | AMC | Avg. |
> | --- | --- | --- | --- | --- | --- | --- | --- |
> | Qwen2.5-Math-7B | 43.9 ± 2.5 | 9.1 ± 2.0 | 14.5 ± 1.5 | 11.1 ± 2.0 | 4.7 ± 1.3 | 31.4 ± 1.8 | 19.1 ± 1.9 |
> | Qwen2.5-Math-7B-Instruct | 81.0 ± 2.0 | 36.6 ± 3.3 | 40.1 ± 2.1 | 10.9 ± 2.0 | 9.1 ± 1.8 | 48.3 ± 1.9 | 37.7 ± 2.2 |
> | PRIME | 81.7 ± 2.0 | **40.0 ± 3.4** | 40.9 ± 2.1 | 18.1 ± 2.4 | 14.4 ± 2.2 | 55.9 ± 1.9 | 41.8 ± 2.3 |
> | Oat-Zero | 79.4 ± 2.0 | 35.7 ± 3.3 | 43.4 ± 2.2 | **32.9 ± 3.0** | 10.1 ± 1.9 | 59.4 ± 1.9 | 43.5 ± 2.4 |
> | Simple-RL-Zero | 75.5 ± 2.2 | 26.7 ± 3.0 | 35.2 ± 2.1 | 26.6 ± 2.8 | 8.3 ± 1.7 | 55.6 ± 1.9 | 38.0 ± 2.3 |
> | Open-Reasoner | 82.6 ± 1.9 | 35.3 ± 3.3 | 46.6 ± 2.2 | 17.0 ± 2.4 | 13.2 ± 2.1 | 52.0 ± 1.9 | 41.1 ± 2.3 |
> | On-Policy | 84.5 ± 1.8 | 38.2 ± 3.3 | 47.2 ± 2.2 | 22.2 ± 2.6 | 14.2 ± 2.2 | 62.1 ± 1.8 | 44.7 ± 2.3 |
> | SFT | 82.1 ± 1.9 | 38.7 ± 3.3 | 42.0 ± 2.1 | 23.0 ± 2.7 | 22.3 ± 2.6 | 53.5 ± 1.9 | 43.6 ± 2.4 |
> | SFT + RL | 86.5 ± 1.7 | 39.8 ± 3.4 | 48.3 ± 2.2 | 29.4 ± 2.9 | 22.3 ± 2.6 | 62.2 ± 1.8 | 48.1 ± 2.4 |
> | LUFFY | **87.3 ± 1.7** | 38.7 ± 3.3 | **55.9 ± 2.2** | 29.1 ± 2.9 | **24.0 ± 2.7** | **64.7 ± 1.8** | **50.0 ± 2.4** |
>
> - We will update our next version with these results.
>
> **Q5: The legend in Figure 6 is not clear. Although it can be understood that dotted lines represent entropy, it should be labelled. Line 296: typo as “uently”**
>
> - Thanks for your advice and reminder. We will fix these issues in our next version.
>
> We hope our responses answer your questions, and again, thanks for your advice to improve our paper!

---

> > ### Comment · Reviewer_VkPb · 2025-08-05
> >
> > Thanks for the feedback. In light of feedback and other reviews, i would like to maintain my current score.

---

### Decision · Program_Chairs · 2025-09-17

**Decision:**

Accept (poster)

**Comment:**

The idea is nice and after the author response it was as consensus accept with 2 enthusiastic reviews.